# Learning to Reason Iteratively and Parallelly for Complex Visual Reasoning Scenarios

**Shantanu Jaiswal**[1,2]   **Debaditya Roy**[2]   **Basura Fernando**[2,3]   **Cheston Tan**[2,3]

[1] Carnegie Mellon University [2] IHPC, A*STAR Singapore
[3] Centre for Frontier AI Research, A*STAR Singapore
Correspondence to: `sjaiswa3@cs.cmu.edu`

## Abstract

Complex visual reasoning and question answering (VQA) is a challenging task that requires compositional multi-step processing and higher-level reasoning capabilities beyond the immediate recognition and localization of objects and events. Here, we introduce a fully neural *Iterative* and *Parallel Reasoning Mechanism* (IPRM) that combines two distinct forms of computation – iterative and parallel – to better address complex VQA scenarios. Specifically, IPRM's *"iterative"* computation facilitates compositional step-by-step reasoning for scenarios wherein individual operations need to be computed, stored, and recalled dynamically (e.g. when computing the query *"determine the color of pen to the left of the child in red t-shirt sitting at the white table"*). Meanwhile, its *"parallel"* computation allows for the simultaneous exploration of different reasoning paths and benefits more robust and efficient execution of operations that are mutually independent (e.g. when counting individual colors for the query: *"determine the maximum occurring color amongst all t-shirts"*). We design IPRM as a lightweight and fully-differentiable neural module that can be conveniently applied to both transformer and non-transformer vision-language backbones. It notably outperforms prior task-specific methods and transformer-based attention modules across various image and video VQA benchmarks testing distinct complex reasoning capabilities such as compositional spatiotemporal reasoning (AGQA), situational reasoning (STAR), multi-hop reasoning generalization (CLEVR-Humans) and causal event linking (CLEVRER-Humans). Further, IPRM's internal computations can be visualized across reasoning steps, aiding interpretability and diagnosis of its errors. Source code at: `https://github.com/shantanuj/IPRM_Iterative_and_Parallel_Reasoning_Mechanism`

## 1 Introduction

Visual reasoning and question answering (VQA) at its core requires a model to identify relevant visual operations, execute those operations, and then combine their results to make an inference. Complex visual reasoning scenarios (depicted in fig. 1) are particularly challenging in this regard. They require models to reason compositionally over a large number of reasoning steps and to engage in a variety of higher-level reasoning operations such as causal linking, logical reasoning, and spatiotemporal processing that extend beyond core perception capabilities. In this context, two powerful computational priors exist – iterative and parallel. While each has its own limitations, when combined, they can complement each other and effectively address the challenges of complex VQA tasks. Specifically, **iterative computation**, wherein individual operations are identified and composed in a step-by-step manner, is a beneficial prior for multi-step reasoning scenarios explored by past VQA works [28, 25, 18]. However, pure iterative computation can exhibit limitations in

38th Conference on Neural Information Processing Systems (NeurIPS 2024).

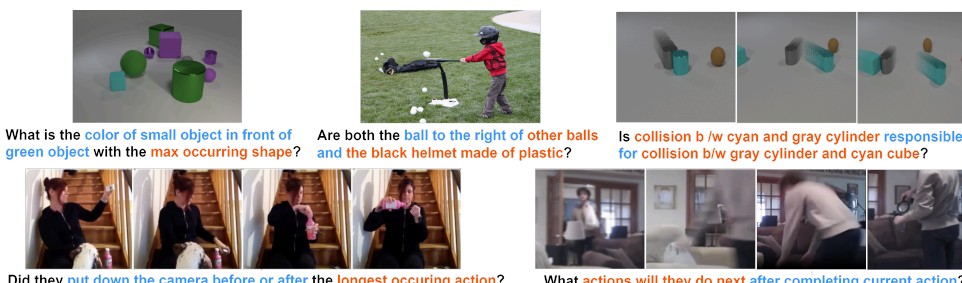

Figure 1: Complex VQA scenarios (CLEVR-Humans [35], GQA [29], CLEVRER-Humans[51]), AGQA[20] and STAR[79]) wherein combination of iterative (step-by-step) computation (blue phrases) and parallel computation (orange phrases) can be beneficial for reasoning.

scenarios wherein the entailed visual operations are independent of one-another, or where distinct stimuli need to be processed and tracked simultaneously.

For example, consider the first scenario shown in fig. 1. When executing the language phrase *"maximum occurring shape"* (i.e. *"what shape appears the most"*), a purely iterative method would: (i) compute the count of each shape (each of which itself could take multiple iterations), (ii) then update and maintain the counts in memory (without forgetting count of all previous shapes), and (iii) finally, recall each shape's count to compute the *"maximum"* [1]. Besides taking more reasoning steps than required, such computation also increases the demand for information retention and recall in memory, which in this scenario could scale by the number of shapes to be counted. In complex video reasoning scenarios, a purely iterative method would similarly struggle in tracking and reasoning over multiple co-occurring events. In such scenarios, from both efficiency and efficacy perspectives, it is advantageous to process operations or stimuli in parallel, rather than solely iteratively.

More generally, **parallel computation** facilitates the simultaneous maintenance and exploration of distinct reasoning paths, and thereby enables reasoning to be more comprehensive, efficient and robust. For example, to compute *"maximum occuring shape"*, parallel compute can enable distinct shape queries to be simultaneously computed prior to computing the *"maximum"* operation. Similarly, it is effective for other scenarios illustrated in fig. 1 such as in processing independent logical clauses (*"{are X} and {Y made of plastic}"*) or when tracking and processing co-occurring events in videos.

Such computation can be implicitly realized in conventional transformer-based parallel attention mechanisms [70]. However, transformer-based attention does not explicitly incorporate iterative compositional computation [14, 41], which as described is beneficial for multi-step reasoning scenarios wherein operations need to be composed sequentially. Accordingly, while parallel computation may effectively compute the result of *"maximum occurring shape"* in fig. 1, it would potentially struggle to integrate the result with further operations such as *"green object with .."*, *"small object in front of green .."*, and *"color of .."* that need to be computed step-by-step to answer the question.

Based on the above insights, we design the **Iterative and Parallel Reasoning Mechanism (IPRM),** a novel neural reasoning architecture that combines step-by-step iterative computation with the ability to process multiple independent operations and stimuli simultaneously. Inspired by how humans utilize working memory [17, 9] to facilitate complex reasoning, IPRM internally maintains a latent memory of parallel "operation states", keyed to which are "results states". Given vision and language inputs, IPRM performs the following *iterative* computation. First, it forms a new set of *parallel* operations by retrieving relevant language information conditioned on its prior operation states. Then, it "executes" these operations *parallelly* by retrieving relevant visual information conditioned on its new operations as well as prior result states. Finally, it integrates its new operations (and their results) into memory by dynamically composing these operations with one-another as well as prior operation states, and subsequently, repeats the entire process in its next *iterative* step.

This strategy effectively enables us to take advantage of both parallel and iterative computations and notably helps improve state-of-arts across various complex image and video reasoning tasks using a single reasoning mechanism. Equally importantly, IPRM's internal computations can be visualized

---

[1]Assuming *"max"* is applied after counts of all shapes are computed.

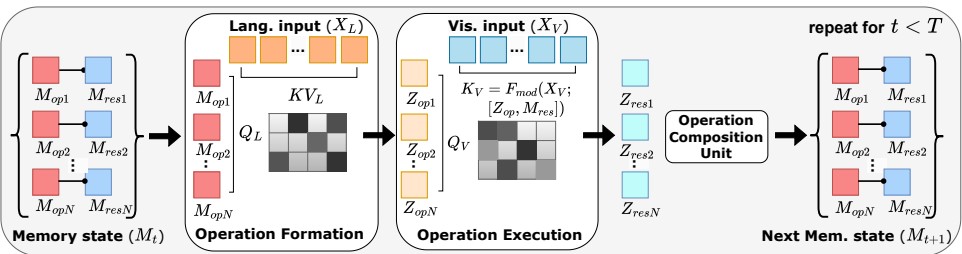

Figure 2: IPRM's computation flow diagram. First, a new set of N-parallel latent operations $\mathbf{Z_{op}}$ are retrieved from language features $\mathbf{X_L}$ conditioned on prior operation states $\mathbf{M_{op}}$. Then, visual features $\mathbf{X_V}$ are queried conditioned on both $\mathbf{Z_{op}}$ and prior result states results $\mathbf{M_{res}}$, to form the new results $\mathbf{Z_{res}}$. Finally, both $\mathbf{Z_{res}}$ and $\mathbf{Z_{op}}$ are passed to the Operation Composition Unit (see 2.3), the output of which becomes the new memory state $\mathbf{M}$.

across reasoning steps, which helps better interpret what operations it was doing and accordingly where it was looking visually when processing a complex reasoning scenario.

## 2 Iterative and Parallel Reasoning Mechanism

Our proposed iterative-and parallel-reasoning mechanism (IPRM) is a fully-differentiable neural architecture. Given visual features $\mathbf{X_V} \in \mathbb{R}^{N_V \times D_V}$ and language or task-description features $\mathbf{X_L} \in \mathbb{R}^{N_L \times D_L}$, IPRM outputs a *"reasoning result"* $\mathbf{y_s} \in \mathbb{R}^{D_m}$ and, optionally, a set of *"reasoning result tokens"* $\mathbf{Y_R} \in \mathbb{R}^{N_m \times D_m}$. As previously mentioned, IPRM operates iteratively for $T$ reasoning steps and internally, maintains an explicit memory $\mathbf{M} : \{\mathbf{M_{op}}, \mathbf{M_{res}}\}$. The memory is modelled as a set of *"operation states"* $\mathbf{M_{op}} \in \mathbb{R}^{N_{op} \times D_m}$, keyed to which are *"result states"* $\mathbf{M_{res}} \in \mathbb{R}^{N_{op} \times D_m}$ as shown in fig. 2. Here, $N_{op}$ denotes the number of parallel operations to be computed while $D_m$ denotes the mechanism's internal feature dimension. On a high level, at each reasoning step (denoted by $t \in \{1, \cdots, T\}$), IPRM performs the following computations:

1. First, conditioned on the existing operation states $\mathbf{M_{op,t}}$, we retrieve relevant information from language or task-description features $\mathbf{X_L}$ to form a new set of latent operations $\mathbf{Z_{op,t}} \in \mathbb{R}^{N_{op} \times D_m}$. We term this computation as *"Operation Formation"*.

$$\mathbf{Z_{op,t}} = \mathbf{Op\_Form}(\mathbf{X_L}; \mathbf{M_{op,t}}) \tag{1}$$

2. Then, conditioned on the latent operations $\mathbf{Z_{op,t}}$ and the existing result state $\mathbf{M_{res,t}}$, we attend and retrieve relevant information from visual features $\mathbf{X_V}$ which represents a new set of latent results $\mathbf{Z_{res,t}} \in \mathbb{R}^{N_{op} \times D_m}$ corresponding to $\mathbf{Z_{op,t}}$. We term this computation as *"Operation Execution"*.

$$\mathbf{Z_{res,t}} = \mathbf{Op\_Exec}(\mathbf{X_V}; [\mathbf{Z_{op,t}}, \mathbf{M_{res,t}}]) \tag{2}$$

3. Finally, to facilitate interaction amongst parallel operations, we perform inter-operation attention. Here, each operation $\mathbf{Z_{op\_k,t}}; \mathbf{k} \in \{1, \cdots, N_{op}\}$, is composed with other operations in $\mathbf{Z_{op,t}}$ as well as prior operation states $\mathbf{M_{op[t-W:t]}}$ within a lookback-window $W$. The corresponding result $\mathbf{Z_{res\_k,t}}$ is similarly composed with other results $\mathbf{Z_{res,t}}$ and prior result states denoted as $\mathbf{M_{res[(t-W):t]}}$. We term this computation as *"Operation Composition"*

$$\mathbf{M_{t+1}} = \mathbf{Op\_Comp}(\{\mathbf{Z_{op,t}}, \mathbf{Z_{res,t}}\}, \mathbf{M_{[(t-W):t]}}) \tag{3}$$

As shown in eq. (3), this output is the new memory state $\mathbf{M_{t+1}} : \{\mathbf{M_{op}}, \mathbf{M_{res}}\}$.

The overall computation flow is illustrated in fig. 2, and we provide specific details and intuitions behind these computations in the following sub-sections.

## 2.1 Operation Formation

The *"operation formation"* stage conceptually models a reasoner that based on its prior set of operations, decides what language features to retrieve in order to form the next set of relevant operations. This can be effectively implemented through conventional attention mechanisms. Specifically, the cumulative set of prior operations (maintained in $\mathbf{M_{op,t}}$) can be projected to form the 'query' $\mathbf{Q_{L,t}} \in \mathbb{R}^{N_{op} \times D_m}$ representing "what features to look for". The language features $\mathbf{X_L}$ can be projected to form the 'key' $\mathbf{K_L} \in \mathbb{R}^{N_L \times D_m}$ and 'value' $\mathbf{V_L} \in \mathbb{R}^{N_L \times D_m}$. Finally, the new set of latent operations $\mathbf{Z_{op,t}}$ can be retrieved by computing $\texttt{attn}(\mathbf{Q_L}, \mathbf{K_L}, \mathbf{V_L})$. These steps are formally represented below:

$$\mathbf{Q_{L,t}} = \mathbf{W_{L,q2}}(\texttt{Tanh}(\mathbf{W_{L,q1}}(\mathbf{M_{op,t}}))), \mathbf{K_L} = \mathbf{W_{L,k}}(\mathbf{X_L}), \mathbf{V_L} = \mathbf{W_{L,v}}(\mathbf{X_L}) \tag{4}$$

$$\mathbf{Z_{op,t}} = \texttt{attn}(\mathbf{Q_{L,t}}, \mathbf{K_L}, \mathbf{V_L}) \tag{5}$$

Here, $\mathbf{W_{L,q2}} \in \mathbb{R}^{D_m \times D_m}$, $\mathbf{W_{L,q1}} \in \mathbb{R}^{D_m \times D_m}$, $\mathbf{W_{L,k}} \in \mathbb{R}^{D_m \times D_l}$ and $\mathbf{W_{L,v}} \in \mathbb{R}^{D_m \times D_l}$. Note $\mathbf{K_L}$ and $\mathbf{V_L}$ are not computation-step dependent and only computed once. We use a simple linear-modulated formulation (with appropriate broadcasting and projection weight $\mathbf{W_a} \in \mathbb{R}^{D_k \times 1}$) to implement $\texttt{attn}(.)$ (further details in appendix sec. 13).

## 2.2 Operation Execution

In the *"operation execution"* stage, the reasoner determines what visual features need to be retrieved depending on both the newly formed operations and existing result states. To model the constituent visual attention mechanism, we draw insights from existing recurrent visual reasoning methods [28, 69] that incorporate feature modulation for memory-guided attention. Specifically, we retrieve a set of feature modulation weights $\mathbf{S_{V,t}} \in \mathbb{R}^{N_{op} \times D_m/r}$ through a joint projection of the new operations $\mathbf{Z_{op,t}}$ and prior results $\mathbf{M_{res,t}}$ as shown in eq. (6).

$$\mathbf{S_{V,t}} = \mathbf{W_{V,s}}([\mathbf{W_{V,op}}(\mathbf{Z_{op,t}}), \mathbf{W_{V,res}}(\mathbf{M_{res,t}})]) \tag{6}$$

Here, $r$ is a feature reduction ratio [23, 31]. $\mathbf{S_{V,t}}$ is then applied dimension wise to a projection of $\mathbf{X_V}$ to retrieve an intermediate attention key $\mathbf{K'_{V,t}} \in \mathbb{R}^{N_{op} \times N_k \times D_m/r}$. The final attention key $\mathbf{K_{V,t}}$ is then obtained through a joint multi-layer-projection of $\mathbf{K'_{V,t}}$ and the previously projected representation of $\mathbf{X_V}$ as shown in eq. (7).

$$\mathbf{K'_{V,t}} = \mathbf{S_{V,t}} \odot \mathbf{W_{V,k1}}(\mathbf{X_V}), \ \mathbf{K_{V,t}} = \mathbf{W_{V,k3}}(\phi(\mathbf{W_{V,k2}}([\mathbf{W_{V,k1}}(\mathbf{X_V}), \mathbf{K'_{V,t}}]))) \tag{7}$$

Finally, the attention query and value are formed through separate projections of $\mathbf{Z_{op,t}}$ and $\mathbf{X_V}$ respectively. These are then fed together with $\mathbf{K_{V,t}}$ to the attention function to retrieve the new operation results $\mathbf{Z_{res,t}}$ as shown in eq. (8). Intuitively, the overall process allows for both prior results and the new set of operations to jointly guide visual attention.

$$\mathbf{Q_{V,t}}, \mathbf{V_{V,t}} = \mathbf{W_{V,q}}(\mathbf{Z_{op,t}}), \mathbf{W_{V,v}}(\mathbf{X_V}), \mathbf{Z_{res,t}} = \texttt{attn}(\mathbf{Q_{V,t}}, \mathbf{K_{V,t}}, \mathbf{V_{V,t}}) \tag{8}$$

Here, $\mathbf{W_{V,op}} \in \mathbb{R}^{D_m/r \times D_m}$, $\mathbf{W_{V,res}} \in \mathbb{R}^{D_m/r \times D_m}$, $\mathbf{W_{V,s}} \in \mathbb{R}^{D_m/r \times 2D_m/r}$, $\mathbf{W_{V,k1}} \in \mathbb{R}^{D_m/r \times D_v}$, $\mathbf{W_{V,k2}} \in \mathbb{R}^{D_m/r \times 2D_m/r}$, $\mathbf{W_{V,k3}} \in \mathbb{R}^{D_m/r \times D_m/r}$, $\mathbf{W_{V,q}} \in \mathbb{R}^{D_m/r \times D_m}$ and $\mathbf{W_{V,v}} \in \mathbb{R}^{D_m \times D_m}$.

## 2.3 Operation Composition

Finally, in the *"operation composition"* stage, the reasoner first integrates the executed operations $\mathbf{Z_{op,t}}$ and their results $\mathbf{Z_{res,t}}$ into the existing memory state $\mathbf{M_t}$ through a simple recurrent update as shown in eqs. (9) and (10). Then, to mitigate redundancy amongst parallel operations and to retrieve relevant knowledge from prior-step operations, it dynamically composes individual operation states $\mathbf{M'_{op,t+1}}$ with other operation states in $\mathbf{M'_{op,t+1}}$ and also prior operation states in $\mathbf{M_{op,t-W:t}}$. Here, $W$ is an attention look-back window.

This composition is achieved through computing inter-operation attention as illustrated in fig. 3. Specifically, $\mathbf{M'_{op,t+1}}$ is projected to obtain a set of queries $\mathbf{Q_{op,t}}$, while the token-wise concatenation of $\mathbf{M'_{op,t+1}}$ and $\mathbf{M_{op,t-W:t}}$ are projected to obtain the operation attention keys $\mathbf{K_{op,t}}$ and values $\mathbf{V_{op,t}}$. A second set of values $\mathbf{V_{res,t}}$ are also formed through projection of respective result states

as shown in eq. (14). Further, an identity attention mask $\mathbf{I_{N_{op}}}$ is used to ensure that operations in $\mathbf{Q_{op,t}}$, can only attend to other operations and not themselves. This is done to enable a higher degree of operation composition. As shown in eq. (15), $\mathbf{Q_{op,t}}$, $\mathbf{K_{op,t}}$, $\mathbf{V_{op,t}}$ and $\mathbf{I_{N_{op}}}$ are passed to the attention operation, which outputs an intermediate representation $\mathbf{M''_{op,t+1}}$ and the softmaxed-attention weights $\mathbf{A_{op,t}}$. $\mathbf{M''_{op,t+1}}$ is then added to a projection of $\mathbf{M'_{op,t+1}}$ to effectively combine attended operation states with the original operation states, and thereby form the next mem. operation state $\mathbf{M_{op,t+1}}$.

Finally, the next result states are obtained by applying $\mathbf{A_{op,t}}$ on $\mathbf{V_{res,t}}$ and then adding a projection of $\mathbf{M'_{res,t+1}}$ as shown in eq. (17). Note $\mathbf{A_{op,t}}$ is specifically utilized to ensure that results are composed based on attentions between operation states. Here, all the mentioned weights $\mathbf{W_{..}} \in \mathbb{R}^{D_m \times D_m}$ and $[\cdot;\cdot]$ represents token-wise concatenation. $\mathbf{I_{N_{op}}}$ in eq. (15) is an identity matrix which is concatenated with zeros (i.e. unmasked) for window tokens if window len. W > 0.

$$\mathbf{M'_{op,t+1}} = \mathbf{W_{opU}}(\mathbf{Z_{op,t}}) + \mathbf{W_{opH}}(\mathbf{M_{op,t}}) \tag{9}$$

$$\mathbf{M'_{res,t+1}} = \mathbf{W_{resU}}(\mathbf{Z_{res,t}}) + \mathbf{W_{resH}}(\mathbf{M_{res,t}}) \tag{10}$$

$$\mathbf{Q_{op,t}} = \mathbf{W_{op,q}}(\mathbf{M'_{op,t+1}}) \tag{11}$$

$$\mathbf{K_{op,t}} = \mathbf{W_{op,k}}([\mathbf{M'_{op,t+1}}; \mathbf{M_{op,t-W:t}}]) \tag{12}$$

$$\mathbf{V_{op,t}} = \mathbf{W_{op,v}}([\mathbf{M'_{op,t+1}}; \mathbf{M_{op,t-W:t}}]) \tag{13}$$

$$\mathbf{V_{res,t}} = \mathbf{W_{res,v}}([\mathbf{M'_{res,t+1}}; \mathbf{M_{res,t-W:t}}]) \tag{14}$$

$$\mathbf{M''_{op,t+1}}, \mathbf{A_{op,t}} = \texttt{attn}(\mathbf{Q_{op,t}}, \mathbf{K_{op,t}}, \mathbf{V_{op,t}}, \text{mask}=\mathbf{I_{N_{op}}}) \tag{15}$$

$$\mathbf{M_{op,t+1}} = \mathbf{M''_{op,t+1}} + \mathbf{W_{op,u2}}(\mathbf{M'_{op,t+1}}) \tag{16}$$

$$\mathbf{M_{res,t+1}} = \mathbf{A_{op,t}}(\mathbf{V_{res,t}}) + \mathbf{W_{res,v2}}(\mathbf{M'_{res,t+1}}) \tag{17}$$

**Obtaining Reasoning Summary** As mentioned before, our proposed mechanism outputs a set of *"reasoning result tokens"* $\mathbf{Y_R}$ and a *"reasoning result"* $\mathbf{y_s}$. $\mathbf{Y_R}$ is simply equivalent to the last memory result states $\mathbf{M_{res,T+1}}$. To obtain $\mathbf{y_s}$, we perform attention on the last operation states $\mathbf{M_{op,T+1}}$ by utilizing a summary representation $\mathbf{l_s} \in \mathbb{R}^{D_l}$ of $\mathbf{X_L}$ as the attention-query.

We set $\mathbf{l_s}$ to be the first token in case of transformer-based language backbones and as last hidden state in case of LSTM-based language backbones. As shown in eq. (18), $\mathbf{l_s}$ is projected to obtain a single-token attention query $\mathbf{p_q}$ while $\mathbf{M_{op,T+1}}$ is projected to obtain the attention keys $\mathbf{P_k}$. The attention value is simply the result states $\mathbf{M_{res,T+1}}$, and the output of the attention function is the *"reasoning result"*. Intuitively, this computation corresponds to the reasoner deciding which final operation states in $\mathbf{M_{op,T+1}}$ are most relevant to the summary of the input language or task-description $\mathbf{X_L}$, based on which corresponding result states $\mathbf{M_{res,T+1}}$ are weighted and retrieved.

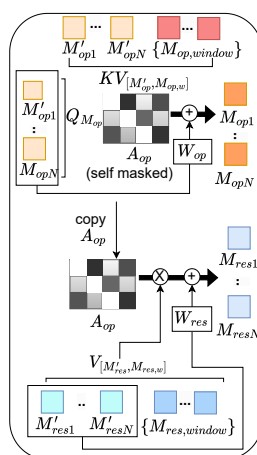

$$\mathbf{p_q}, \mathbf{P_k} = \mathbf{W_{pq,q}}(\mathbf{l_s}), \mathbf{W_{pk,k}}(\mathbf{M_{op,T+1}}) \tag{18}$$

$$\mathbf{y_s} = \texttt{attn}(\mathbf{p_q}, \mathbf{P_k}, \mathbf{M_{res,T+1}}) \tag{19}$$

Figure 3: Operation Composition Unit

Here, $\mathbf{W_{pq,q}} \in \mathbb{R}^{D_m \times D_l}$ and $\mathbf{W_{pk,k}} \in \mathbb{R}^{D_m \times D_m}$.

**Reasoning mechanism general applicability.** Our proposed iterative and parallel reasoning mechanism is an end-to-end trainable neural module. It can be conveniently applied on top of different vision and language backbones, and be trained directly as a new computational block with no specific adjustments. Further, IPRM is weight-tied which means that its number of parameters is constant regardless of number of computation steps and parallel operations. We provide parameter and computational details along with module implementation in appendix sec. C.

Table 1: Comparison of IPRM with videoQA methods on STAR (left) and AGQAv2 (right). All methods operate on 32 frames unless otherwise mentioned in (). *Not directly compared as utilizes additional surrogate tasks / benchmarks and num. of frames not reported.

| Model | Int. | Seq. | Pred. | Feas. | Avg. |
|---|---|---|---|---|---|
| LRR*[5] | 73.7 | 71.0 | 71.3 | 65.1 | 70.3 |
| LRR (w/o surrogate) | 54.5 | 48.7 | 44.3 | 45.5 | 48.2 |
| All-in-One [72] | 47.5 | 50.8 | 47.7 | 44.0 | 47.5 |
| Temp[ATP][7] | 50.6 | 52.8 | 49.3 | 40.6 | 48.3 |
| MIST [18] | 55.5 | 54.2 | 54.2 | 44.4 | 51.1 |
| InternVideo (8) [75] | 62.7 | 65.6 | 54.9 | 51.9 | 58.7 |
| SeViLA-BLIP2 [86] | 63.7 | 70.4 | 63.1 | **62.4** | 64.9 |
| Concat-Att-4L | 68.1 | 71.4 | 66.6 | 55.2 | 65.3 |
| Cross-Att-4L | 67.5 | 72.1 | 64.4 | 58.5 | 65.6 |
| IPRM | **71.8** | **77.7** | **71.0** | 59.1 | **69.9** |

| Metric | HCRN[39] | AIO[72] | Temp[7] | MIST[18] | GF [4] | IPRM |
|---|---|---|---|---|---|---|
| obj-rel | 40.3 | 48.3 | 50.2 | 51.7 | 55.0 | **57.8** |
| superlative | 33.6 | 37.5 | 39.8 | 42.1 | 44.6 | **48.0** |
| sequencing | 49.7 | 49.6 | 48.3 | 67.2 | 53.2 | **75.6** |
| exist | 50.0 | 50.8 | 51.8 | 60.3 | 59.1 | **62.4** |
| duration | 43.8 | 45.4 | 49.6 | **54.6** | 52.8 | 50.7 |
| act. recog. | 5.5 | 19.0 | 19.0 | 19.7 | 14.2 | **20.0** |
| open | 36.3 | - | - | 50.6 | 56.1 | **58.6** |
| binary | 48.0 | - | - | 58.3 | 54.2 | **62.3** |
| all | 42.1 | 48.6 | 49.8 | 54.4 | 55.1 | **60.4** |

# 3 Experiments

We evaluate IPRM on STAR[79], AGQAv2[20] and CLEVRER-Humans[51] for video reasoning tasks and CLEVR-Humans[35], GQA [29] and CLEVR-CoGenT [34] for image reasoning tasks. For all tasks, we set IPRM's parallel operations ($N_{op}$) to 6, reasoning steps ($T$) to 9, reduction ratio ($r$) to 2 and window length ($W$) to 2 (informed by ablative analysis detailed in sec. 3.3). We follow task-specific practices (detailed in appendix C.1) for respective vision and language backbones in our primary experiments, besides also demonstrating integration of IPRM with large-scale VL backbones such as CLIP. Further, besides task-specific methods, we also consider two prominent transformer-based VL modules as baselines – concat-att (where lang. and vis. tokens are concatenated as in [36, 37]) and cross-att (where lang. tokens are "query" to vis. tokens as "key" and "value"; as in [43, 1]). Further implementation and training details are provided in appendix sec.C.

## 3.1 Video Reasoning and Question Answering (STAR, AGQAv2 and CLEVRER-Humans)

We first evaluate IPRM on recent video reasoning benchmarks. STAR [79] and AGQAv2 comprise real-world videos and test multiple reasoning skills in context of situational reasoning and compositional spatiotemporal reasoning respectively. STAR contains 60K questions testing four broad types of video reasoning abilities: *feasibility*, *interaction*, *prediction* and *sequence*. Meanwhile, AGQAv2 contains 2.27M balanced questions distributed across 16 different question types. As shown in table 1, IPRM obtains 69.9% average acc. on STAR and 60.4% overall acc. on AGQAv2, outperforming prior videoQA-specific methods by  5% on both benchmarks.

Interestingly, on STAR IPRM obtains an 8% and 7% improvement over SeViLA-BLIP2 on the predictive and sequencing scenarios respectively. This is possibly due to IPRM's capability to reason over multiple events simultaneously across frames, which may enhance its capacity to retrieve and cumulatively reason on relevant information needed to predict future events and determine appropriate sequences. However, IPRM performs less effectively than SeViLA-BLIP2 in feasibility scenarios, possibly because these scenarios require not only visual reasoning but also commonsense knowledge, which can benefit from integration with larger-scale vision-language backbones and auxiliary training tasks such as introduced in LRR [5].

Similarly, On AGQAv2, IPRM improves performances across various question types, notably achieving a 8% improvement in questions that require determining sequence of events. Further, IPRM also outperforms both 4-layer concat- and cross-attention modules on STAR (scaling further attention-layers was not found to benefit performance as detailed in appendix table 7).

Next, we evaluate IPRM on the CLEVRER-Humans benchmark [51], which comprises synthetic videos of simultaneous object motions and multiple collisions, and tests a model's ability to determine causal links between events. We perform zero-shot, finetuned and from-scratch evaluation. As shown in table 2, IPRM outpeforms task-specific neurosymbolic models NS-DR and VR-DP as well as state-of-the-art ALOE across the three settings. Specifically, IPRM improves zero-shot per-question acc. by 7%, finetuned per-question acc. by 18.8% and scratch per-question acc. by 6.2%. These results further suggest that IPRM can better track and process co-occuring events, and in this case, more accurately determine causal links.

Table 2: Comparison of methods for CLEVRER-Humans [51] (Opt. is per option acc. and Qs. is per question acc.). IPRM achieves state-of-art across settings.

| Model | Zero-shot | | Finetune | | Scratch | |
|---|---|---|---|---|---|---|
| | Opt. | Qs. | Opt. | Qs. | Opt. | Qs. |
| NS-DR[84] | 51.0 | 32.0 | - | - | - | - |
| VRDP[13] | 50.9 | 31.6 | - | - | - | - |
| CNNLSTM[51] | 50.3 | 30.0 | 51.7 | 34.2 | 51.5 | 30.8 |
| CNNBERT[51] | 52.9 | 32.0 | 52.0 | 30.2 | 50.1 | 30.4 |
| ALOE[12] | 54.0 | 26.9 | 51.8 | 31.7 | 52.7 | 32.1 |
| IPRM | **61.7** | **38.9** | **74.1** | **53.0** | **62.0** | **38.3** |

Table 3: Comparison of methods on CLEVR-Humans (zero-shot and finetuned setting), CLEVR-CoGenT (ValA is in-domain; ValB is out-of-domain setting) and CLOSURE. IPRM performs strongly without requiring extra supervision.

| Model | Extra supv. | CLV-Hum | | CLV-CoGen | | CLOSURE |
|---|---|---|---|---|---|---|
| | | ZS | FT | ValA | ValB | ZS Avg. |
| PG+EE [35] | Programs | 54.0 | 66.6 | 96.6 | 73.7 | 75.6 |
| NS-VQA [85] | Programs | - | 67.8 | **99.8** | 63.9 | **77.2** |
| FiLM [60] | None | 56.6 | 75.9 | 98.3 | 78.8 | 56.9 |
| MAC [28] | None | 57.4 | 81.5 | 99.0 | 78.3 | 73.8 |
| MDETR [36] | Bound. Box | 59.9 | 81.7 | **99.8** | 76.7 | - |
| IPRM | None | **63.8** | **85.5** | 99.1 | **80.3** | 75.6 |

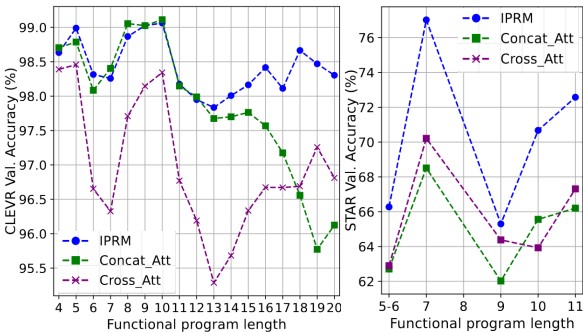

Figure 4: Acc. of IPRM (blue) across program lengths for CLEVR (left) and STAR (right). IPRM has significantly higher accs. at longer program lengths.

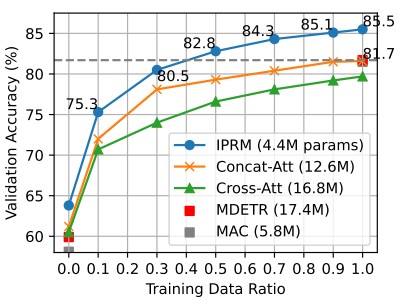

Figure 5: IPRM performance on CLEVR-Humans at different training data ratios of Cross- and Concat-Att.

## 3.2 Compositional Image Reasoning (CLEVR-Humans, CLEVR-CoGen and GQA)

Here, we evaluate IPRM on challenging compositional image reasoning benchmarks. CLEVR-Humans tests generalization of multi-hop reasoning to free-form human crowdsourced questions which entail reasoning skills/scenarios beyond a model's training on the original CLEVR dataset and provides a limited finetuning set (2.5% of CLEVR). Similarly, CLEVR-CoGenT tests generalization on novel attribute compositions not observed in training (e.g."*gray cubes*" and "*red cylinders*" are in training but eval. is on "*gray cylinders*" and "*red cubes*"; see suppl. for exact specification). The CLOSURE [3] benchmark further tests systematic generalization for different question type compositions.

As shown in table 3, IPRM achieves 3.9% and 3.8% improvements in zero-shot and fully-finetuned performance over prior state-of-art vision-language model MDETR. Further, IPRM neither requires bounding-box pre-training supervision (as done in MDETR) nor functional programs, and can be trained directly with only vision-language inputs and task supervision. fig. 5 illustrates IPRM's performance across different training ratios compared to MDETR and cross- and concat-att transformer modules. Notably, IPRM exceeds MDETR's fully-finetuned performance by 1.1% with only half of training data. Further, IPRM exhibits these improvements while being relatively lightweight (4.4M params) in comparison to MDETR's transformer blocks (17.4M; ∼4x more parameters) and cross- and concat-VL attention methods (16.8M and 12.6M respectively). These results suggest that IPRM exhibits strong generalization capabilities and more sample-efficient learning of novel reasoning skills and scenarios in context of multi-step imageQA.

For CLEVR-CoGenT, IPRM achieves state-of-art results in out-of-domain generalization on novel attribute compositions and outperforms MDETR (having parallel transformer compute) by 3.6% and MAC (iterative method) by 2%. This suggests the combination of iterative and parallel computation as done in IPRM can implicitly enable more disentangled feature processing and thereby improve compositional learning of primitive attributes in context of multi-step imageQA. Similarly, on CLOSURE, IPRM achieves an average zero-shot accuracy of 75.6% which is highest amongst fully neural reasoning methods (that require no extra supervision) and is close to the neurosymbolic method NS-VQA which utilizes ground truth programs and object bounding boxes supervision. Further, detailed breakdown of models on CLOSURE question types is provided in appendix table 6.

Table 4: Performance comparison on GQA with imageQA methods and large-scale models that do not utilize ground-truth scene graphs. * indicates large-scale pretrained VL model. **Utilizes ground truth scene graphs, programs and bounding boxes for auxiliary training.

| | LCGN [25] | MCAN[87] | LXMERT*[66] | 12-in-1*[49] | OSCAR*[46] | CFR** [55] | IPRM |
|---|---|---|---|---|---|---|---|
| GQA | 55.8 | 57.4 | 60.0 | 60.0 | **61.6** | 72.1 | 60.3 |

We also evaluate IPRM on the GQA benchmark which tests compositional VQA on real-world images. We perform comparisons with prior VQA methods as well as large-scale VL models such as LXMERT and 12-in-1 that do not utilize the ground truth scene graphs in GQA. As shown in table 4, IPRM achieves 60.3% , outperforming prior reasoning methods such as MCAN and LCGN as well as large-scale VL models such as LXMERT and 12-in-1. However, IPRM's performance is behind the larger VL model OSCAR which performs pretraining on 4.3M data samples collated from COCO, VG, SBU and Flickr. Note, IPRM is a standalone reasoning module only trained on GQA balanced with no pre-training. Further, IPRM when trained with perfect perception (i.e. ground truth object bounding boxes and attributes), achieves 87.2% on GQA validation set suggesting strong reasoning performances can be achieved through advancements in visual detectors and backbones. Finally, in appendix B.3, we demonstrate that IPRM more effectively enhances the performance of frozen CLIP variants on complex reasoning benchmarks compared to scaling traditional cross- and concat-attention modules.

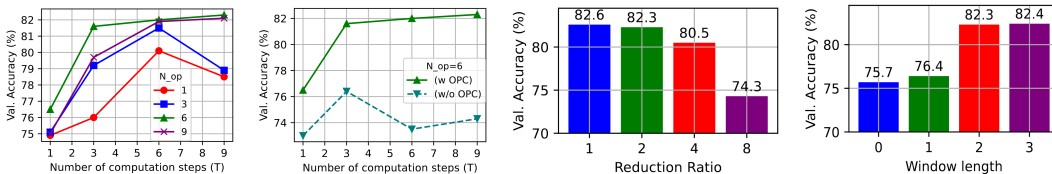

Figure 6: IPRM Model ablations in order: **(i)** Impact of number of parallel operations ($N_{op}$) vs computation steps ($T$). **(ii)** Impact of Operation Composition Block (OPC). **(iii)**: Impact of reduction ratio ($r$) and **(iv)** memory window length ($W$).

## 3.3 Model ablations and reasoning visualization

We perform ablations to study the contributions of IPRM's salient components. First, we analyze the impact of varying number of parallel operations ($N_{op}$) against number of iterative computation steps ($T$). We compare models with $N_{op} \in \{1, 3, 6, 9\}$ and $T \in \{1, 3, 6, 9\}$, resulting in 16 different models. Given the large amount of ablative models, we perform analysis on a reduced-resolution setting of CLEVR-Humans with pretraining on CLEVR for 15 epochs. As shown in fig. 6 (plot (i)), we find that $T$ and $N_{op}$ appear to be co-dependent and that neither by itself can lead to high performance. E.g. setting $T = 1$ generally results in peformances around 75% regardless of $N_{op}$, while setting $N_{op}= 1$ or 3 results in a sharp performance drop of 3% when changing $T$ to 9 from 6. In contrast, for $N_{op} > 3$, we observe that performance increases steadily with higher $T$, suggesting that a higher number of parallel operations may prevent overfitting in a model with high computation steps. Overall, we find that ($N_{op} = 6, T = 9$) and ($N_{op} = 9, T = 9$) are the two best performing models achieving above 82% accuracy (with the former preferred as $N_{op} = 6$ performs better for diff. $T$ compared to $N_{op} = 9$).

Next, we study the impact of the operation composition block (OPC) by evaluating $N_{op} = 6$ (and diff. computation steps $T$) with and without OPC. As shown in fig. 6 (plot (ii)), while $T = 1$ has a relatively low drop of ∼2%, the performance drops are more significant for higher $T$. The ($N_{op} = 6$, $T = 9$) model which reached ∼82% acc. with OPC, drops to ∼74% acc. without OPC.

Finally, we study the impacts of the dimension reduction ratio ($r$) and memory-lookback window length ($W$). As shown in fig. 6 (plot (iii)), $r = 2$ leads to negligible drop (0.3%) in performance compared to no dimension reduction (i.e. $r = 1$) while being computationally faster and requiring less memory. However, setting $r$ to 8 significantly deteriorates performance. Similarly, as shown in fig. 6 (plot (iv)), a setting of window-length $W$ is 2 works sufficiently well, and decreasing it below that significantly impacts performance.

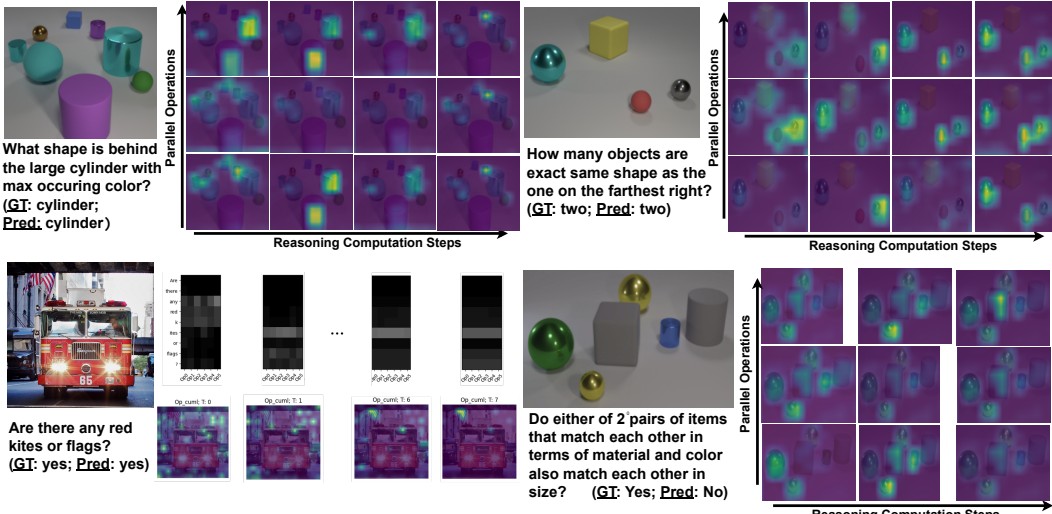

Figure 7: **Condensed reasoning visualization of IPRM**. In the top two examples, IPRM correctly utilizes both parallel and iterative computation to arrive at the correct answer. The bottom left example shows IPRM's cumulative lang. and visual attentions when solving a real-world GQA example. The bottom right example, shows an error case where IPRM seems to misunderstand question and outputs wrong ans. with less relevant attentions. See appendix for further reasoning visualizations and error cases, and supplemental for further CLIP integrated visualizations on GQA examples.

In fig. 4, we assess the performance of IPRM across different functional program lengths (a proxy for reasoning steps) on the CLEVR and STAR benchmarks. As shown, IPRM maintains a high performance across both short and longer reasoning steps and for both image and video VQA scenarios. We also provide a condensed illustration of IPRM's reasoning visualization in fig. 7. Here, for brevity, we only show the model's visual attention for a subset of parallel operations and computation steps (see suppl. for full visualization). The top left example ("what shape .. max occuring color") illustrates the model's usage of both iterative and parallel computation. At the first reasoning step, the model appears to be doing parallel operations to both identify "large cylinder" and compute "max occuring color". In the next step, it appears to have found "max occuring color" (cyan) and seems to check which of the two large cylinders match the color. Next, it highlights the correct cylinder and in the final step, it locates and predicts the correct object behind. The top right example in fig. 7 similarly shows computation for a question involving spatial ("farthest right"), similarity ("same shape") and counting ("how many") operations. The bottom left example illustrates IPRM's simultaneous language and visual attentions updated across reasoning steps when solving a real world (GQA) example. Finally, the bottom right example is an error-case where IPRM seems to incorrectly understand or execute requisite operations, thereby not producing expected attentions for "items with matching material and color" throughout steps.

## 4 Related Work

**Visual reasoning methods and vision-language models.** Multiple prior works have introduced effective visual reasoning methods in context of image and video question answering [35, 52, 2, 85, 15, 50, 69, 56, 32, 54, 53, 76]. Prominent works include NMN[2], FILM [60], NSM [27], MAC [28], MCAN [87], NS-VQA [85], ALOE [12], VR-DP [13], SHG-VQA [68], MIST [18] and OCRA [76]. In contrast to these works that show applicability of methods for particular VQA benchmarks/tasks, our work explores a more general direction of integrating parallel and iterative computation in a reasoning framework that we show to be effective for multiple complex VQA benchmarks and reasoning scenarios. More recently, vision-language models [44, 73, 37, 66, 45, 46, 74] and multimodal large-language-models [57, 47] with transformer-based mechanisms have shown impressive reasoning capabilities at scale. Notable examples include CLIP [61], GPT [57], Gemini [67], MDETR [36], LXMERT [66], VinVL [88], BLIP [44, 43], Flamingo [1], LlavA [47], BEiT [74] and All-in-One [72]. We believe our work is complimentary to these developments, as it

contributes an alternative and possibly more effective reasoning mechanism that can be integrated with such models in future to enhance complex VQA capabilities.

**Memory and recurrence-augmented transformers.** Multiple works have identified limitations of purely feedforward computation as realized in transformers and worked on encoding recurrence [30, 26, 11, 71] and memory-augmented computation [80, 8]. Notably, Recurrent Memory Transformer [8] and MemFormer[81] introduce recurrent and dynamic memory to improve language modelling capabilities. More recently, EMAT [82] introduces efficient memory to augment knowledge retrieval, MemViT [80] introduces a cache memory to retain prior context for long-video tasks, and [71] introduces memory for more effective action anticipation. While these methods study recurrent and memory-augmented computation on specific natural language processing and computer vision tasks, our work focuses on the integration of iterative-parallel computation and working memory in a single neural reasoning mechanism beneficial for complex VQA scenarios.

## 5  Conclusion

We introduced a novel fully-differentiable and end-to-end trainable iterative and parallel reasoning mechanism (IPRM) to address complex VQA scenarios. We comprehensively evaluated IPRM on various complex image and video VQA benchmarks testing distinct reasoning capabilities, and found it improves state-of-arts on multiple such benchmarks. We also performed quantiative ablations to study individual impacts of parallel and iterative computation besides qualitative analysis of IPRM's reasoning computation visualization.

## 6  Limitations and Future work

Here, we note possible limitations of IPRM. Similar to existing VQA and deep-learning methods, IPRM may reflect biases that are present in the training distribution of VQA benchmarks. This may lead it to overfit to certain image inputs or question forms and possibly provide skewed answers in such scenarios. Further, the utilized vision-language backbones in our experiments may also entail visual, language and cultural biases in their original training distribution which may permeate to IPRM upon integration for VQA scenarios. In this regard, we hope the capability to visualize intermediate reasoning of IPRM and diagnose its error cases (as shown in section 3.3) can serve a useful tool to benefit interpretability in VQA and identify possible reasoning biases that may emerge in the model.

For future work, scaling the IPRM architecture to a foundational video-language model by integrating it with large-scale transformer-based vision-language models and relevant instruction-tuning approaches presents an exciting research opportunity. Moreover, while we designed and evaluated IPRM in the context of complex VQA, we believe it has the potential to operate as a general reasoning mechanism applicable to tasks beyond visual reasoning and question answering, such as language processing and embodied reasoning.

**Acknowledgment** This research/project is supported by the National Research Foundation, Singapore, under its NRF Fellowship (Award# NRF-NRFF14-2022-0001). This research is also supported by funding allocation to B.F. and C.T. by the Agency for Science, Technology and Research (A*STAR) under its SERC Central Research Fund (CRF), as well as its Centre for Frontier AI Research (CFAR).

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

# A    Appendix / supplemental material

# B    Elaborated Experiments and Results Discussion

## B.1    Further comparisons on CLEVR-Humans, CLEVR-CoGenT, CLOSURE and NLVRv1

Here, we provide further comparisons with benchmark-specific methods for CLEVR-Humans [35], CLEVR-CoGenT [34], CLOSURE [3] and NLVRv1 [63] (not reported in main paper due to space limitations). As mentioned in main paper, these benchmarks utilize synthetic images and are a test of pure visual reasoning capabilities that are minimally influenced by increased world knowledge or usage of stronger visual backbones.

CLEVR-Humans as already mentioned in main paper evaluates a model's reasoning generalization capabilities to unseen scenarios or question forms. CLEVR-CoGenT studies compositional attribute generalization. Specifically, it has two conditions – i) cond.A wherein all cubes have color $\in \{gray, blue, brown, yellow\}$ and cylinders $\in \{red, green, purple, cyan\}$ (spheres can be any color), and ii) cond.B wherein color-sets are switched b/w cubes and cylinders. A model is then trained on one condition and evaluated on both the original and alternate condition. A higher accuracy on the alternate condition indicates that the model learns more 'compositionally' as it generalizes better to novel shape-color combinations with less feature/attribute combination overfitting.

Table 5: Elaborated results on CLEVR-Humans (left), CLEVR-CoGenT (middle) and NLVRv1 (right). IPRM achieves state-of-art across the three benchmarks and does not require additional supervision such as bounding boxes or functional programs. * requires func. programs supervision / pre-defined dataset-specific neural modules. ▼ requires object bounding-boxes supervision.

| Model | CLV-Hum ZS | CLV-Hum FT |
|---|---|---|
| PG+EE* [35] | 54.0 | 66.6 |
| NS-VQA▼* [85] | - | 67.8 |
| RAMEN [62] | 57.8 | - |
| FiLM [60] | 56.6 | 75.9 |
| GLT [6] | - | 75.8 |
| LEFT [22] | - | 78.8 |
| MAC [28] | 57.4 | 81.5 |
| MDETR▼ [36] | 59.9 | 81.7 |
| IPRM | **63.8** | **85.5** |

| Model | CoGenTr-A ValA | CoGenTr-A ValB | CoGenFT-B ValA | CoGenFT-B ValB |
|---|---|---|---|---|
| NS-VQA▼* [85] | **99.8** | 63.9 | - | - |
| MDETR▼ [36] | **99.8** | 76.7 | - | - |
| StackAtt-MLP[83] | 80.3 | 68.7 | 75.7 | 75.8 |
| PG + EE* [34] | 96.6 | 73.7 | 76.1 | 92.7 |
| Tbd-Net* [52] | 98.8 | 75.4 | 96.9 | 96.3 |
| MAC [28] | 99.0 | 78.3 | 97.2 | 96.1 |
| FILM [60] | 98.3 | 78.8 | 81.1 | 96.9 |
| IPRM | 99.1 | **80.3** | **98.0** | **98.2** |

| Model | NLVR1 Test-U |
|---|---|
| CNN-RNN [63] | 56.3 |
| MAC [28] | 59.4 |
| FILM [60] | 61.2 |
| NMN* [2] | 62.0 |
| N2NMN* [24] | 66.0 |
| CNN-BiATT [65] | 66.1 |
| IPRM (scratch) | 63.8 |
| IPRM-CLV-FT | **73.0** |

Finally, NLVRv1 evaluates language-grounded visual reasoning. Each sample of this benchmark comprises a set of three synthetic images and a composite natural language statement about the images which can evaluate to True or False and requires various visual-linguistic reasoning skills.

As shown in table 5, IPRM achieves state-of-art results across the three benchmarks and does not require pre-annotated bounding-boxes or functional programs as additional supervision. For **CLEVR-Humans** (table 5 left), it outperforms larger-scale models such as MDETR and RAMEN in zero-shot performance even though the latter is pre-trained on multiple VQA datasets. It also increases state-of-art in finetuned setting by 3.8%.

For **CLEVR-CogenT** (table 5 centre) , IPRM achieves the highest generalization results amongst methods in both the CoGen-Train A and Finetune B. Specifically, it obtains 80.3% acc. on cond. B (when trained on cond. A), which is 1.5% higher than the previous state-of-art cond.B method FILM and 3.6% higher than MDETR. When further finetuned on cond.B, IPRM generalizes for both cond.A and cond.B achieving 98.0% and 98.2% unlike FILM which overfits to cond.B and thereby has poor performance on cond.A. Further, its performance on cond.A (99.1%) is highest amongst methods that do not utilize bounding box or localization supervision and marginally lower than MDETR and NS-VQA (which utilize bounding-box supervision).

**For NLVRv1** (table 5 right), IPRM model trained from scratch achieves 63.8% acc. and performs competitively with existing task-specific state-of-art model CNN-BiAtt. When finetuned from its CLEVR checkpoint, we find IPRM achieves 73.0% acc. which is 7% higher than existing visual inputs state-of-art for NLVRv1 and suggests strong reasoning transfer capabilities of IPRM. It further outperforms the N2NMN method which requires pre-defined neural modules to be identified for the dataset.

Finally, we have also provided results of models on CLOSURE for different subsets / question types in table 6. For the CLEVR-environment, we have visualized some failure cases in figs. 10 and 11. More cases can be run through the visualization framework in provided source code.

Table 6: Zero-shot performances of models on CLOSURE subsets (2 trials run for IPRM)

| Model (ZeroShot) | and_mat_spa | embed_mat_spa | embed_spa_mat | or_mat | or_mat_spa | compare_mat | compare_mat_spa |
|---|---|---|---|---|---|---|---|
| FiLM | 41.4±18 | 62±11 | 93.4±2 | 34.4±11 | 35.2±5.5 | 66.2±8.5 | 65.8±5 |
| MAC | 63.7±25 | 76.8±11 | 99±0.15 | 75.2±13 | 70.4±15 | 65.3±8.6 | 66.2±6.4 |
| NS-VQA* | 33.3±18 | 97.3±4.6 | 98.3±5 | 69.5±28 | 50.4±27 | 96.1±6.2 | 95.4±6.7 |
| PG-Tensor-NMN* | 37.6±17 | 79.8±1.4 | 95.6±5.6 | 34.1±8.1 | 25.5±11 | 89.2±2.3 | 89.8±2.7 |
| PG-Vector-NMN* | 32.5±17 | 97.3±2.4 | 97.2±3.9 | 64.9±25 | 47.4±23 | 95.3±5.7 | 94.4±6.4 |
| IPRM | 83.1±6.4 | 80.3±7.2 | 99.2±0.1 | 75.7±9.4 | 67.1±8.1 | 62.2±7.3 | 61.5±5.8 |

## B.2 Elaborated STAR results and ablations for video reasoning tasks

We provide results on the STAR Test, further baselines and model ablations for video reasoning tasks in table 7.

Table 7: **Left:** Results on STAR [79] official hidden test set (evaluation server) with ground-truth vision (GT V) and predicted vision (PR V); **Right:** Results on STAR val. set with num. of sampled frames =32 unless otherwise stated in ().

| Model | Setup | STAR-Test | | | | |
|---|---|---|---|---|---|---|
| | | Int. | Seq. | Pred. | Feas. | Avg. |
| Vis-BERT[45] | GT V | 34.7 | 35.9 | 31.2 | 31.4 | 34.7 |
| CLIP-BERT[40] | GT V | 36.3 | 38.9 | 30.7 | 29.8 | 36.5 |
| NS-SR[79] | GT V | 42.6 | 46.3 | 43.4 | 43.9 | 44.5 |
| IPRM | GT V | **79.1** | **85.8** | **82.0** | **71.4** | **79.6** |
| Vis-BERT[45] | - | 33.6 | 37.2 | 31.0 | 30.8 | 34.8 |
| CLIP-BERT[40] | - | 39.8 | 43.6 | 32.2 | 31.4 | 36.7 |
| NS-SR[79] | PR V | 30.9 | 31.8 | 30.2 | 29.7 | 30.7 |
| SHG-VQA [68] | - | 48.0 | 42.0 | 35.3 | 32.5 | 39.5 |
| GF [4] | - | 56.1 | 61.3 | 52.7 | 45.7 | 53.9 |
| mPLUG [42] | - | 60.4 | 65.6 | 57.5 | 49.6 | 58.3 |
| IPRM | PR V | **72.0** | **78.1** | **71.5** | **59.4** | **70.3** |

| Model | Int. | Seq. | Pred. | Feas. | Avg. |
|---|---|---|---|---|---|
| All-in-One [72] | 47.5 | 50.8 | 47.7 | 44.0 | 47.5 |
| Temp[ATP][7] | 50.6 | 52.8 | 49.3 | 40.6 | 48.3 |
| MIST [18] | 55.5 | 54.2 | 54.2 | 44.4 | 51.1 |
| InternVideo (8) [75] | 62.7 | 65.6 | 54.9 | 51.9 | 58.7 |
| SeViLA-BLIP2 [86] | 63.7 | 70.4 | 63.1 | **62.4** | 64.9 |
| Concat-Att-2L | 66.0 | 68.9 | 66.4 | 55.1 | 64.1 |
| Concat-Att-4L | 68.1 | 71.4 | 66.6 | 55.2 | 65.3 |
| Cross-Att-4L | 67.5 | 72.1 | 64.4 | 58.5 | 65.6 |
| Concat-Att-6L | 66.3 | 71.4 | 66.6 | 55.7 | 65.0 |
| Cross-Att-6L | 59.8 | 63.0 | 56.4 | 49.9 | 57.3 |
| IPRM(m1,t1) | 65.6 | 69.8 | 65.1 | 54.5 | 63.8 |
| IPRM(m1, t9) | 70.0 | 75.7 | 70.2 | 57.8 | 68.4 |
| IPRM(m6, t1) | 69.5 | 75.2 | 68.6 | 57.4 | 67.7 |
| IPRM | **71.8** | **77.7** | **71.0** | 59.1 | **69.9** |
| IPRM (GT V) | 78.7 | 85.5 | 81.7 | 71.2 | 79.3 |

## B.3 CLIP Integration Results

We provide results with frozen CLIP [61] visual backbones including CLIP VIT-L/14, CLIP VIT-B/16 and CLIP VIT-L/14@336px on GQA [29], NLVRv2 [64] and CLEVR-Humans in table 8. We empirically found utilizing a Distil-roberta language backbone to be more effective than the associated CLIP language backbone, and hence used the former for question processing. We compare with alternate prominent vision-language attention mechanisms including Cross-att and Concat-att blocks as well as a simple joint projection of vision and language pooled representations (referred as Wt-Proj-Att). As shown in the table, IPRM can enhance performance for the CLIP variants across GQA, NLVRv2 and CLV-Humans in comparison to concat and cross-att blocks. Further, it is more parameter efficient with only 5.5M additional parameters in comparison to 4-layer as well as 2-layer stacks of Cross-Att (9.2M 2-layer, 17.6M 4-layer) and Concat-Att (7.2M 2-layer, 13.6M 4-layer). With regards to computational FLOPs, IPRM consumes 5.9GFLOPs which is marginally higher than Cross-Att 4-layer config (3.1GFLOPs) and lower than Concat-Att 4-layer config (8.9GFLOPs). Note, that the performance benefits of adding further layers of cross- or concat-att blocks are observed to be minimal after 4 layers, and can also depend on the amount of training data available. E.g. Both cross- and concat-att blocks of 2 layers had better performances on NLVRv2 (which has a limited set of training questions relative to GQA and CLEVR) in comparison to 4 layer config.

## B.4 Further reasoning computation visualizations

We provide elaborate reasoning computation visualizations of IPRM showing the lang. and vis. attentions across parallel operations and computation steps during *operation formation* and *operation execution* stages. Fig. 8 shows a scenario wherein IPRM correctly utilizes parallel and iterative computations to compute intermediate operations of "find object close to front", "retrieve/compare

Table 8: **Left:** Comparison of IPRM with prominent vision-language attention mechanisms with CLIP VIT-L/14 backbones on CLEVR-Humans, GQA and NLVRv2 benchmarks ('4L' indicates 4 att layers; 'x' indicates model did not converge). **Right:** Results with other CLIP variants VIT-B and VIT-L@ 336 on GQA and NLVRv2.

| Model (CLIP VIT-L/14 bbone) | +Param | +GFLOPs | GQA TestD | NLVR2 Test | CLV-H ZS | CLV-H FT |
|---|---|---|---|---|---|---|
| Wt-Proj-Fusion | 0.6M | 0.1 | 53.5 | 60.8 | 58.5 | 74.4 |
| Cross-Att (2L) | 9.2M | 1.5 | 55.1 | 62.1 | - | - |
| Concat-Att (2L) | 7.2M | 4.4 | 55.3 | 60.5 | - | - |
| Cross-Att (4L) | 17.6M | 3.1 | 57.4 | 54.4 | 60.3 | 80.0 |
| Concat-Att (4L) | 13.6M | 8.9 | 58.7 | 55.9 | 61.2 | 81.1 |
| Cross-Att (6L) | 26.0M | 4.5 | 56.8 | x | 60.8 | 80.4 |
| Concat-Att (6L) | 19.7M | 13.3 | 57.4 | x | 62.0 | 81.8 |
| IPRM | 5.2M | 5.9 | **59.2** | **65.1** | **64.3** | **84.6** |

| Model (CLIP VIT-B/16 bbone) | GQA TestD | NLVR2 Test |
|---|---|---|
| Wt-Proj-Fusion | 51.4 | 59.9 |
| Cross-Att | 54.6 | 56.6 |
| Concat-Att | 56.0 | 57.4 |
| IPRM | 55.9 | 60.8 |

| Model (CLIP VIT-L/14@336 bbone) | GQA TestD | NLVR2 Test |
|---|---|---|
| Wt-Proj-Fusion | 54.0 | 61.1 |
| Cross-Att | 57.4 | 58.4 |
| Concat-Att | 57.3 | 59.1 |
| IPRM | 59.0 | 65.3 |

shape and size", "find applicable objects with both same shape and size". Fig. 9 shows another correct prediction of IPRM, and this time, its intermediate reasoning visualization is useful to determine that the entailed reasoning appears sensible. Fig. 10 shows an incorrect prediction by IPRM and its intermediate reasoning visualizations also suggest that IPRM did not understand the question and thereby did not attend to relevant objects. Finally, Fig. 11 shows a scenario wherein while IPRM produces the correct answer, it's intermediate reasoning appears imprecise which makes the prediction (and underlying reasoning) less reliable. We provide further visualizations with a CLIP VIT-L/14 backbone on GQA samples in the supplemental jupyter notebook output (html format for easier viewing).

## C   Model implementation and experiment details

We implement IPRM in PyTorch [58] as a generic vision-language module receiving a set of input vision (or scene-representation) tokens and input language (or task-representation) tokens. We provide **Python-style pseudocode of IPRM in figs 12, 13 and 14**. For all experiments, we set the internal dimension of IPRM to 512 and use the same configuration of num. parallel operations ($N_{op}$)=6, num. computation steps (T)=9, reduction ratio (r)=2 and window size (W)=2. We follow benchmark-specific conventions for vision-language backbones that are detailed below in sec. C.1. For CLIP [61], we utilize the official models from Huggingface [77]. All experiments are performed on a single NVIDIA A40 GPU with 46GB memory and averaged over 3 trials with different random seeds wherever possible (done for primary experiments on STAR, AGQA, CLEVRER-Humans, CLEVR-Humans). Unless otherwise specified, the learning rate is initialized to 1e-4 with Adam [38] optimizer and gradient clipping value of 8. The learning-rate is reduced based on validation acc. plateau with reduction factor 0.5, threshold 0.001 and patience 0. Further experiment hyper-parameters and settings are provided below. Source code at: https://github.com/shantanuj/IPRM_Iterative_and_Parallel_Reasoning_Mechanism.

### C.1   Benchmark-specific experiment details

**CLEVR-Humans**. We use the CLEVR-Humans dataset from [35] which comprises images from original CLEVR dataset [34] and human crowdsourced questions. We use a batch size of 216 for training. We use the same language encoder (Distil-Roberta[48] from Huggingface[78]) as in existing state-of-art MDETR [36] and frozen ResNet101 backbone layer 3 spatial features (as in [28, 52, 35]). We perform all ablation experiments with 14x14x1024 visual features. Each ablation model is first pretrained for 10 epochs on the original CLEVR dataset (the initial learning rate for IPRM is 1e-4 and for language encoder is 1e-5) and then finetuned on CLEVR-Humans for 40 epochs with early stopping (learning rate of 1e-4 throughout). As observed in prior work [52], we similarly found in multiple scenarios with occluded objects that visual attention only partially identified such objects. Hence, we simply resampled (bilinear sampling) visual input to obtain 16x16x1024 features and empirically found more complete visual attentions with a corresponding 1.1% improvement in accuracy. The final two best performing model configurations (*Nop=6, T=9, W=2, R=2* and *Nop=6, T=9, W=2, R=1*) from ablations were then pre-trained for 35 epochs on CLEVR and finetuned

on CLEVR-Humans. While we found that configuration *Nop=6, T=9, W=2, R=1* obtains highest zero-shot (ZS) acc. of 65.6% and finetuned (FT) acc. of 86.3%, we adopt *Nop=6, T=9, W=2, R=2* (with 63.3% ZS and 85.4% FT acc.) as our optimal model given its lesser parameters and FLOPs.

**GQA**. We use the GQA compositional real-world image question answering dataset from [29]. Based on prior VQA methods on GQA [29, 25, 45, 33], we utilize pre-extracted bounding-box object proposal features and object label predictions obtained from a pretrained object detector [19, 88]. The bounding box coordinates is normalized to range of 0 to 1 based on the original input image size, and the 4 coordinates are transformed to a distributed representation through a learned nonlinear projection. This representation is concatenated with a learned projection of the predicted object labels (initialized with glove[59] 300dim embeddings) to form the final visual input. We train IPRM for 25 epochs with a batch-size of 192 and same hyperparameters as before. We evaluate the final model on both the test-dev split and the official test evaluation server (`https://eval.ai/featured-challenges/225/evaluation`).

**STAR-VideoQA.** We use the STAR-VideoQA dataset for situational reasoning on real-world videos from [79]. Based on previous videoQA methods [79, 40, 45] for STAR, we utilize object bounding boxes, labels, human pose and human-object relations across frames (note: we do not use the situation hyper-graphs or functional programs). We first perform experiments with the provided ground truth object bounding boxes, labels and human-object relations as well as provided human pose predictions from Alphapose [16] as reported in main paper. Each of these is projected to a distributed representation through learned non-linear projections to obtain object token-wise representations. A further learnable positional embedding for each frame is added to these representations which are then flattened across frames to form the visual input to IPRM. For the language encoder, we found both a simple Bi-LSTM and Distil-roberta language encoder obtain similar performance, and hence choose the simpler Bi-LSTM as the language model. We evaluated models on both 16 sampled frames and 32 sampled frames, and empirically found using 32 frames has ∼1.3% higher performance and used it for our primary experiments. A batch size of 64 was used with learning rate 1e-4 over 20 epochs with early stopping. We evaluated the models on both the validation split and official test evaluation server `https://eval.ai/web/challenges/challenge-page/1325/overview`. For the all-predicted (no ground-truth) visual input setup, similar to [79], we utilize a fasterRCNN [19] object extractor, ST-Trans scene graph extractor [10] and the same Alphapose predictor to obtain predicted object bounding boxes, labels, human-object relations and human poses. We observe a drop of ∼9% in predicted setup similar to observations in [79], suggesting further performance can be achieved through better object and relationship detection backbones.

**AGQAv2**. We use the AGQAv2 [20, 21] benchmark that comprises balanced training and test splits. We followed the same methodology as in STAR for language and visual backbones. Since AGQAv2 has a very large training and validation set, we found 8 epochs to be a sufficient number (with most performance increment observed in the 1st epoch itself).

**CLEVRER-Humans.** We use the CLEVRER-Humans dataset introduced in [51] for temporal, physical and causal video reasoning which comprises videos from the original CLEVRER dataset [84]. Similar to in STAR-VideoQA and neurosymbolic models [84, 85], we utilize a pretrained faster-RCNN based object localization and attribute prediction network from [84]. We again form object-level representations by concatenating learned projections of object-bounding box coordinates and predicted object attributes (i.e. color, shape and material). A frame-level learnable positional embedding is added and object-tokens across frames are flattened to form the final visual input to IPRM. For the language encoder, we used a simple bi-LSTM similar to existing methods. Note we do not use the functional programs or event causal graphs in our model. The batch size was 128 with a learning rate of 8e-5 and every 4 frames sampled (resulting on average 32 sampled frames). We evaluated models in the three setups – from scratch, zero-shot (CLEVRER-pretrained) and finetuned (CLEVRER-pretrained). Since the CLEVRER-Humans dataset is relatively small (comprising only 1076 questions ∼ 8 batches; ∼ 0.5% of original CLEVRER), for scratch training we trained for 250 epochs (with early stopping) while for finetuning we finetuned for 150 epochs (with 35 epochs for original CLEVRER training).

**CLEVR-CoGen.** We use the CLEVR-CoGen dataset from [34] and follow the same setup as in CLEVR-Humans. We use a simpler bi-LSTM language encoder for experiments since the questions are synthetic program-generated unlike in CLEVR-Humans (crowdsourced free-form). We trained our model on condition A for 40 epochs (with early stopping) and used the best cond. A validation

performance model to evaluate generalization performance on cond.B. For finetuning on cond.B we finetuned the best cond.A model for 20 epochs and used the best cond.B validation performance model to also evaluate on cond.A. All other hyperparameters are the same as mentioned for CLEVR-Humans.

**NLVR.** We use the NLVRv1 and NLVRv2 datasets from [63, 64]. NLVRv1 comprises 3 synthetic images and a language statement while NLVRv2 comprises 2 real-world images and a lang. statement. For both datasets, the obtained visual tokens for each images was flattened to obtain the final visual input and an image-wise positional embedding was added to indicate image order. For the language encoder, we used a simple Bi-LSTM.

## D   Potential Negative Impact

In relation to VQA and deep-learning methods in general, the deployment of IPRM in real-world applications without thorough consideration of dataset or training distribution biases, could inadvertently reinforce existing vision, language and cultural biases present in the data, leading to erroneous outcomes or skewed answers. Further, the deployment of VQA methods such as IPRM in sensitive domains such as healthcare or scene/footage analysis could raise ethical concerns, including privacy violations, algorithmic reliability, and the potential for unintended consequences stemming from erroneous or biased predictions.

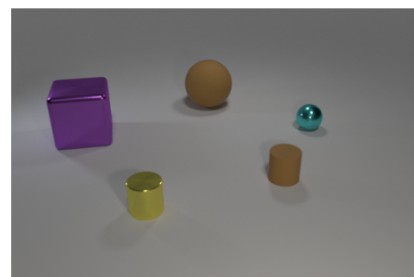

**Are there any objects that have both shape and size in common with the object that is found closest to the front? (GT: Yes; Pred: Yes)**

**Parallel Operations**

**Reasoning Computation Steps**

Figure 8: **Top**: original image and question; **middle**: language attentions across parallel operations (clubbed together; op_k represents parallel operation k) and computation steps. **Bottom**: Visual attentions across parallel ops and computation steps. Here, IPRM correctly utilizes parallel and iterative compute to locate the correct candidate object for prediction (to which all operations attend in last step).

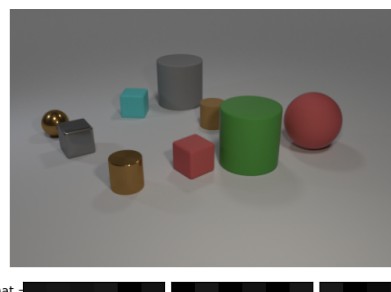

**What shape is the object closest to the gray object with the maximum occuring shape?**
**Pred**: cube **GT**: cube

**Parallel Operations**

**Reasoning Computation Steps**

Figure 9: In this example, IPRM predicts the correct answer and its visual attention trace provides evidence of correct intermediate reasoning. In penultimate reasoning step, IPRM correctly localizes the gray object with maximum occuring shape (cylinder) and in the final step, the parallel operations attend to both the cyan cube and the brown cylinder closest to previously identified gray cylinder.

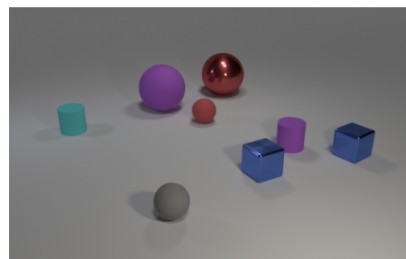

**Are the two objects that are of a primary color, but not red, of the same shape?**
**Pred: no GT: yes**

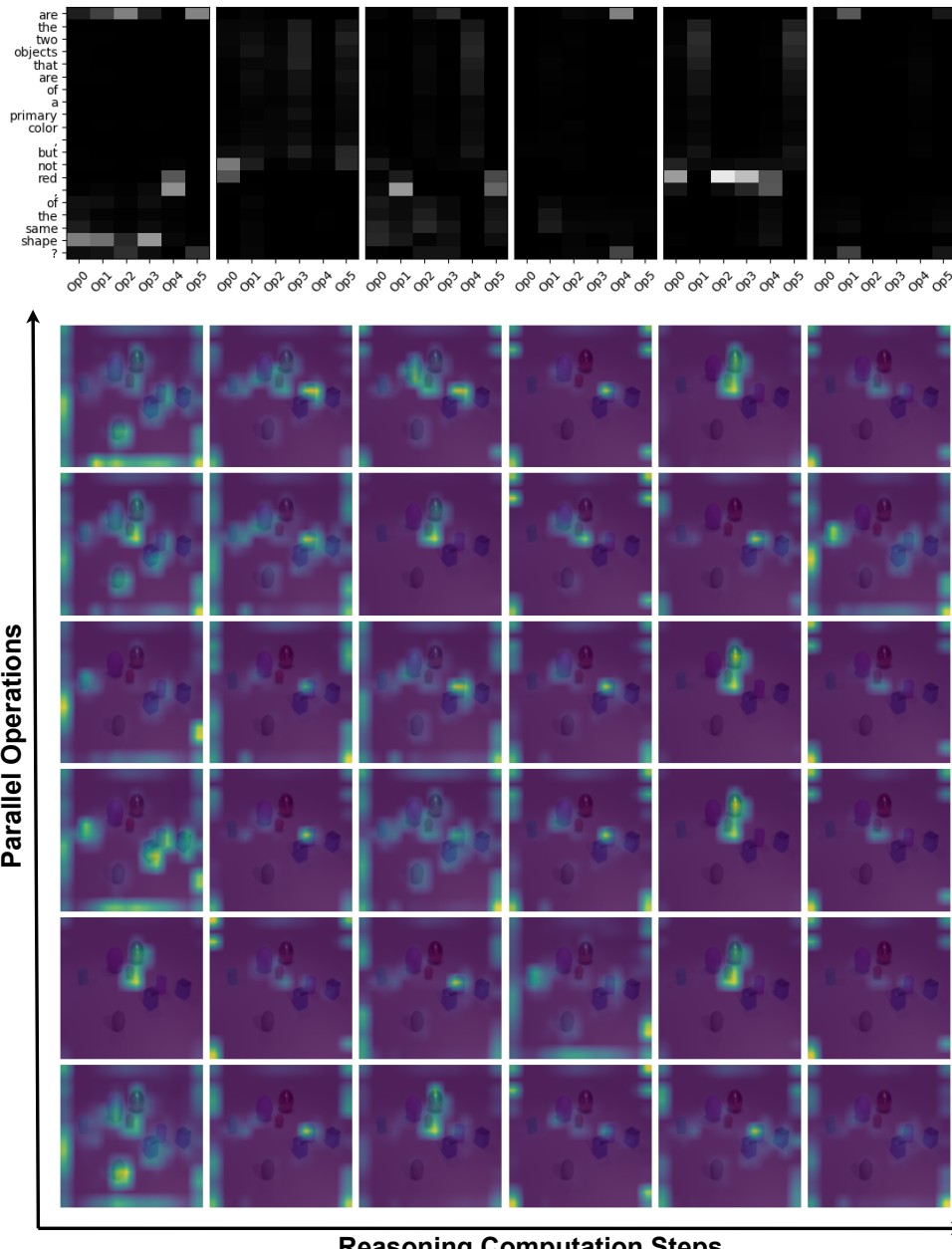

Figure 10: Example where IPRM outputs incorrect answer and the intermediate reasoning appears faulty possibly due to lack of understanding what a "primary color is". The pair of blue (a primary color) cubes in this case should have been identified but are not visually attended in any of the operations across reasoning steps).

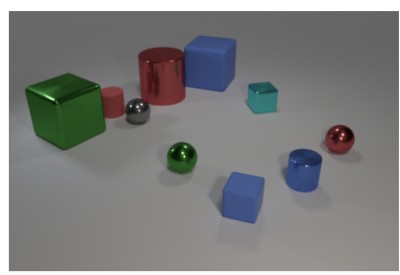

**What shape is the object left of the blue small object with the maximum occuring shape?**
**Pred: sphere GT: sphere**

**Parallel Operations**

**Reasoning Computation Steps**

Figure 11: Example wherein IPRM produces correct answer but its visual attention trace suggests intermediate reasoning may be imprecise. The maximum occuring shape is cube; however both the blue small cylinder and blue small cube appear to be attended in the penultimate step as the "blue small object with max occuring shape" making the reasoning and prediction less reliable.

```
1   def iprm_forward(vis_tokens,      #B×Nv×Dm
2                    lang_tokens,     #B×Nl×Dm
3                    lang_summary_rep,  #B×Dm
4                    num_parallel_ops=6,
5                    num_iterative_steps=9,
6                    mem_window_len=2):
7       mem_op_states = []
8       mem_res_states = []
9       lang_atts = []
10      vis_atts = []
11
12      #0. Initialize memory
13      b, d = vis_tokens.size(0), vis_tokens.size(-1)
14      mem_op_state, mem_res_state = _init_mem_state(num_parallel_ops, b,d)
15      mem_op_states.append(mem_op_state)
16      mem_res_states.append(mem_res_state)
17      for i in range(num_iterative_steps):
18          #1. Form new set of latent operations from lang. token features
19          new_ops, lang_att = operation_formation(lang_tokens, mem_op_state)
20
21          #2. Execute new operations on vis. input to form new results
22          new_ops_results, vis_att = operation_execution(vis_tokens, new_ops,
            ↪   mem_res_state)
23
24          #3. Apply operation composition
25          mem_op_state, mem_res_state = operation_composition(new_ops,
            ↪   new_ops_results, mem_op_states, mem_res_states)
26
27          #4. Maintain memory states within lookback window
28          mem_op_states.append(mem_op_state)
29          mem_res_states.append(mem_res_state)
30          mem_op_states = mem_op_states[min(-1, -mem_window_len):]
31          mem_res_states = mem_res_states[min(-1, -mem_window_len):]
32
33          #5. Store lang. and vis. atts for visualization
34          lang_atts.append(lang_att)
35          vis_atts.append(vis_att)
36
37      #6. 'Pool' final result
38      final_result = pool_final_result(mem_res_state, mem_op_state,
        ↪   lang_summary_rep)
39
40      return final_result, lang_atts, vis_atts
```

Figure 12: IPRM pseudocode (1/3)

```python
#Below, "Lin" refers to a linear layer
#and `"MLP" refers to a 2-layer multi-layer-perceptron layer
def operation_formation(lang_tokens, #B×Nl×Dm
                        prev_op_state #B×Nop×Dm (Nop=num parallel ops)
                        ):
    #1. Form new op "query" based on prior op state
    op_q = MLP_l(prev_op_state) #paper eq. 4

    #2. Use lang_token_feats as attn "key" and "value" (paper eq. 5)
    lang_k = lang_tokens
    lang_v = lang_tokens

    #3. Retrieve new latent ops from lang. rep through attention
    latent_ops, lang_attn = mod_attn(op_q, lang_k, lang_v,
                                     lang_attn_proj) #paper eq.6; L194

    return latent_ops, lang_attn

def operation_execution(vis_tokens, #B×Nv×Dm
                        new_ops, #B×Nop×Dm
                        prev_res_state): #B×Nop×Dm
    #1. Form feature modulation weights (paper eq.7)
    s_v = concat([Lin_op(new_ops), Lin_res(prev_res_state)]) #concat across feat.
    ↪ axis
    s_v = Lin_s(s_v)

    #2. Form visual attention "key" (paper eqs. 8 and 9)
    vis_red_rep =  Lin_v1(vis_tokens)
    mod_vis = s_v * vis_red_rep
    Nop = mod_vis.size(1)
    vis_k = MLP_v(concat([mod_vis, vis_red_rep])) #concat across feat. axis

    #3. Form visual attention "query" and "value" (paper eq. 10)
    vis_q = Lin_op_q(new_ops)
    vis_v = Lin_v2(vis_tokens)

    #4. Obtain new latent "results" through vis attention (paper eq.11)
    latent_results, vis_attn = mod_attn(vis_q, vis_k, vis_v, vis_att_proj)

    return latent_results, vis_attn

def mod_attn(q, k, v, att_proj_layer, attn_mask):
    qk_mult = q*k  #element-wise product
    attn_wt = att_proj_layer(qk_mult) #linear projection (paper L194)
    attn_wt = softmax(attn_wt + (attn_mask * -1e30))
    out = (attn_wt * v).sum() #sum across feature axis
    return out, attn_wt
```

Figure 13: IPRM pseudocode (2/3)

```python
def operation_composition(new_ops, #B×Nop×Dm
                          new_res, #B×Nop×Dm
                          mem_op_states, #list of W elements: B×Nop×Dm
                          mem_res_states #list of W elements: B×Nop×Dm
                          ):
    #1. Integrate new-ops and results into memory (paper eq. 12 and 13)
    inter_op_state = Lin_op_u(new_ops) + Lin_op_h(mem_op_states[-1])
    inter_res_state= Lin_res_u(new_res) + Lin_res_h(mem_res_states[-1])

    #2. Concat operation and result states over memory lookback window
    op_states_windowed = concat([inter_op_state, mem_op_states])
    res_states_windowed = concat([inter_res_state, mem_res_states])

    #3. Form inter-operation queries and keys (paper eq. 14)
    op_queries = Lin_op_q(inter_op_state)
    op_keys = Lin_op_k(op_states_windowed)

    #4. Form inter-operation op values and res values (paper eq. 15-16)
    op_values = Lin_op_v(op_states_windowed)
    res_values = Lin_res_v(res_states_windowed)

    #5. Compute inter-operation attention (paper eq. 17)
    attn_mask = identity_matrix(op_keys.size(1))[:op_queries.size(1)]
    new_op_state, op_attn_wt = mod_attn(op_queries, op_keys, op_values,
    ↪   op_attn_proj, attn_mask)

    #6. Obtain new operation and result states (paper eq. 18-19)
    new_op_state = new_op_state + Lin_op_u2(inter_op_state)
    new_res_state = op_attn_wt*new_res_state + Lin_res_v2(inter_res_state)

    return new_op_state, new_res_state

def _init_mem_state(num_parallel_ops, b):
    #slice specified num parallel ops from initialized params ∼ N(0,1)
    op_init_state = op_init_param[:num_parallel_ops]
    res_init_state= res_init_param[:num_parallel_ops]
    #broadcast batch-wise to get B×N_op×Dm
    return op_init_state.repeat(b,1,1), res_init_state.repeat(b,1,1)

def pool_final_result(res_state, op_state, lang_summary_rep):
    pool_q = Lin_pq(lang_summary_rep)
    pool_k = Lin_pk(op_state)
    return mod_attn(pool_q, pool_k, res_state)
```

Figure 14: IPRM pseudocode (3/3)

