# OpenReview forum: "Learning to Reason Iteratively and Parallelly for Complex Visual Reasoning Scenarios"
_NeurIPS.cc/2024/Conference — NeurIPS 2024 poster_

### Official Review · Reviewer_P4wf · 2024-07-12

**Soundness:** 3
**Presentation:** 3
**Contribution:** 3
**Rating:** 7
**Confidence:** 4

**Summary:**

he paper addresses visual reasoning problems. It introduces a fully neural iterative and parallel reasoning mechanism (IPRM) that combines iterative computation with the ability to perform distinct operations simultaneously. Evaluation is performed on visual reasoning datasets such as CLVER, AGQA, CLEVR-Humans and STAR dataset.

**Strengths:**

* The proposed method is novel and interesting.
* The paper is well written and easy to understand.
* The paper includes in-depth analysis of the proposed method.
* The method shows promising zero-shot performance on CLEVRER-Humans,

**Weaknesses:**

* Generalization Results: The method shows zero-shot results only on CLEVRER-Humans. This is in contract to current SOTA video-language models such as LLaVA-Next or LLaMA-Vid which show zero-shot results across multiple datasets such as MSRVTT to TGIF-QA. Can the proposed model architecture be scaled to create more general purpose models similar to current video-language models?

* Multi-turn QA (related to the above point): The standard transformer architecture allows for multi-turn dialogue, where the user can ask the model multiple questions about a video (e.g. LLaVA-Next or GPT-4V). Can the proposed model architecture be extended to allow for multi-turn dialogue?

* Results on STAR: Recent works such as "Look, Remember and Reason: Grounded reasoning in videos with language models, ICLR 2024" outperform the proposed model.

* Results on GQA:  “Coarse-to-Fine Reasoning for Visual Question Answering, CVPR Workshops 2021” achieves 72.1% compared to 60.5% reported for IPRM. (Note that the use of scene graphs during training should not be a reason not to compare, instead the paper should discuss approaches to close the performance gap).

* The reasoning steps in Figure 7 are not very interpretable. In Figure 7 (left) the model does not seem to attend to the large cylinders in the first step. Similarly, in the Figure 7 (middle) the bottom reasoning chain seems to empty steps where the model does not look at any object. It would also be interesting to see the reasoning chains in case of real-world data such as STAR.

**Questions:**

* Can the proposed architecture be scaled to provide zero-shot results on multiple datasets similar to current SOTA video-language models?
* Can the proposed architecture handle multi-turn conversations?
* The paper should compare/discuss SOTA approaches such as  "Look, Remember and Reason: Grounded reasoning in videos with language models, ICLR 2024" on STAR and “Coarse-to-Fine Reasoning for Visual Question Answering, CVPR Workshops 2021” on GQA.

**Limitations:**

Yes.

---

> ### Author Rebuttal · Authors · 2024-08-07
>
> Thank you for your informative review and insightful questions. We provide point-by-point clarifications below:
>
> **Can proposed architecture be scaled to a video-language model that can provide zero-shot results on multiple video-datasets?**
> Scaling our architecture (IPRM) to a video-language model through integration with large-scale foundation models and instruction-tuning approaches such as done in LLaVA-Next or LLaMA-Vid is indeed an intriguing avenue of research.  Since IPRM receives language and visual token inputs parallelly and is an end-to-end trainable module (similar to conventional transformer blocks), we believe it can be effectively applied on top of transformer-based large-scale foundation models and be directly trained end-to-end with necessary pretraining and instruction-tuning objectives for general video-language models.
> However, we note that this scaling will be a significant research undertaking in itself (in terms of computational resources for pretraining/instruction tuning and constituent experiments). Hence, we believe this will be an exciting avenue of research to comprehensively explore in future work. Currently, for this paper, we believe we have already conducted extensive work in i) designing the IPRM architecture, ii) performing thorough architectural analysis, and iii) showing performance improvements across multiple complex videoQA and imageQA benchmarks (such as STAR, AGQAv2 and CLEVR-Humans), which remain challenging even for finetuned state-of-art vision-language models and reasoning methods (such as SeViLA-BLIP2 and MDETR).
>
> **Can the proposed architecture handle multi-turn conversations?**
> As stated in our previous response, our proposed IPRM architecture receives language and visual token inputs parallelly and is an end-to-end trainable module  similar to conventional transformer blocks. We hence believe IPRM can exhibit multi-turn conversation capabilities observed in transformer blocks when combined with vision-language foundation models and instruction-tuning objectives (such as as in LLaVA-Next and LLaMA-Vid). Additionally, since IPRM explicitly models an internal memory state, it may be particularly advantageous for multi-turn conversational tasks that require the retention and recall of past dialogues or interactions in memory over long periods. We believe this will be another exciting avenue for future work, which we can jointly explore with the aforementioned direction on scaling IPRM  to a large-scale video-language model. We will add these points and necessary discussion to our paper’s section on “Limitations and Future Work” (appendix sec. D), and move the section to the main paper in our updated version.
>
> **Discussion of SOTA approaches such as “Look, Remember and Reason” (LRR) and “Coarse-to-Fine Reasoning” (CFR)**
> Thank you for mentioning these works. We will discuss both these works in related work as well as add them to table 1 and table 4 of our paper. We note that our work’s core contribution is a new neural reasoning architecture, while these works primary contribution is introducing necessary surrogate tasks (in LRR) and coarse-to-fine training strategies (in CFR) to address compositional visual reasoning. As such, we believe our work can complement these methods by incorporating their proposed surrogate tasks and training strategies.
>
> Further, inspired by methods that utilize scene graphs to bridge the performance gap, we performed an experiment using symbolic object labels and attributes from ground truth GQA scene graphs as inputs to IPRM. We interestingly found that in such a setting, our model can achieve 87.2% performance on the GQA validation set resulting in approx. 25% performance improvement over utilizing predicted object visual features. We have similarly found that on STAR, with ground truth scene graphs, our model can achieve 79.7% on the test set (approx. 9% improvement over predicted visual inputs).  These results further indicate that the performance gap on visual reasoning can indeed be closed by obtaining more accurate backbone scene graph models. We will include these results and related discussion in appendix section B.1.
>
> **The reasoning steps in Figure 7 are not very interpretable.**
> - We acknowledge that visualization of reasoning steps may be unclear  at first glance (especially for synthetic datasets such as CLEVR). We have plotted the parallel operations vertically and iterative steps horizontally. In our view, the model does show a correct and sensible reasoning trace across iterative steps.
>
> E.g. In the paper fig. 7 (leftmost scenario), the first parallel operation (top left) does indeed attend to two large cylinders, meanwhile the bottom two parallel operations for that iteration attend generally to different objects in the image (possibly to compute results for "max occuring color"). In the subsequent 2nd iterative step, the model more sharply identifies the two large cylinders across parallel operations, while in the third step it again sharply filters the correct 'large cylinder having the max occuring color' (i.e. blue). Finally in the last step, the model attends to the correct object behind the identified cylinder.
>
> Similarly, in fig. 7 (middle scenario), the bottommost reasoning chain does attend to certain objects in iterative step 2 and 4. We wish to clarify that the the certain cases where there is no visual attention is a reflection of identity operations (that arise due to the inclusion of weighted residuals in memory state update as specified in eqs.15 & 16).
>
>
> **Reasoning chains in case of real-world data**
> We agree that it can be more interesting and perhaps also clearer when visualizing reasoning on real-world data. For this, we had provided visualizations on the GQA dataset in the suppl. jupyter notebook (IPRM_CLIP_Visualization.html) and also in fig.1 of uploaded rebuttal pdf. These examples better illustrate our model's entailed reasoning chains.

---

> > ### Comment · Reviewer_P4wf · 2024-08-12
> > **Great work!**
> >
> > The rebuttal has addressed most of my concerns. I will raise my score.

---

> ### Author Response · Authors · 2024-08-12
> **Thank you for your support!**
>
> Dear reviewer P4wf,
>
> Thank you once again for your constructive suggestions and feedback, and for raising your score.

---

### Official Review · Reviewer_3tYc · 2024-07-12

**Soundness:** 4
**Presentation:** 3
**Contribution:** 4
**Rating:** 6
**Confidence:** 4

**Summary:**

This paper introduces the Iterative and Parallel Reasoning Mechanism (IPRM), a novel neural architecture designed to enhance visual question answering (VQA) by combining iterative and parallel computation. The IPRM aims to address the limitations of existing methods that rely solely on either iterative or parallel processing. The iterative component allows for step-by-step reasoning essential for complex queries, while the parallel component facilitates the simultaneous processing of independent operations, improving efficiency and robustness.

The authors propose a lightweight, fully differentiable module that can be integrated into both transformer and non-transformer vision-language backbones. The IPRM outperforms current state-of-the-art methods across several benchmarks, including AGQA, STAR, CLEVR-Humans, and CLEVRER-Humans, demonstrating its capability to handle various complex reasoning tasks. Additionally, the mechanism’s internal computations can be visualized, enhancing interpretability and error diagnosis.

**Strengths:**

1. The combination of iterative and parallel reasoning within a single framework is novel, addressing specific limitations of existing VQA methods.
2. The IPRM demonstrates superior performance across multiple challenging VQA benchmarks, demonstrated by extensive experimental results.
3. The paper is generally well-structured.

**Weaknesses:**

1. While the paper is well-structured, certain sections, particularly those describing the technical details of the IPRM, can be dense and challenging to follow.
2. The paper could benefit from a more detailed discussion on the scalability of the IPRM, particularly in terms of computational resources and training time.

**Questions:**

1. Can the authors provide more insights into the scalability of IPRM? Specifically, how does the computational complexity of IPRM compare to other state-of-the-art methods?
2. It would be helpful to include an analysis of the sensitivity of IPRM’s performance to its hyperparameters, such as the number of parallel operations and the length of the reasoning steps.

**Limitations:**

The authors acknowledge the limitations of their work, particularly regarding the potential scalability challenges and the need for further evaluation on diverse datasets.

---

> ### Author Rebuttal · Authors · 2024-08-07
>
> Thank you for your informative review and useful suggestions. We provide point-by-point clarifications below:
>
>
> **Technical details can be dense and challenging to follow:**
> To enable easier readability and understanding of the method, We have currently provided pseudocode mapping to each equation in appendix Pg. 30 -32. We will reference the pseudocode in the main paper
>
> **Scalability of IPRM in terms of computational resources used and training time; Comparison with other state-of-art methods**
> As currently mentioned in appendix section C L627 on “Model implementation and experiment details”, we perform all our experiments on a single NVIDIA A40 GPU with 46GB memory and 6 CPU threads/number of workers. We will shift these details to the main paper.
> For further clarification and transparency, we report total training times and peak GPU memory consumption for primary datasets we have considered below:
>
>
> | Dataset | Training time | Peak GPU memory |
> | ---------- | ------------------ | ------------------------- |
> | STAR |        ~ 6 hours |  44GB |
> | AGQA | ~ 14 hours | 43GB |
> | CLEVRER-Humans | ~13 hours | 32GB |
> | CLEVR-Humans | ~10 hours | 33GB |
> | GQA | ~13 hours | 40GB |
>
> Further, in appendix table 7 (left), we have currently reported the GFLOPs and parameter comparison of our architecture IPRM against different scales of baseline concat-attention and cross-attention transformer blocks. IPRM has 5.2M params and 5.9GFLOPs. In comparison, a 4 layer configuration of concat-attention has 13.6M params, 8.9GFLOPs and a 4 layer configuration of cross-att has 17.6M params, 3.1 GLOPs. Hence, IPRM exhibits higher parameter efficiency than both concat-attention and cross-attention transformer blocks while consuming a comparable number of GFLOPs.
>
> Regarding computational comparison with existing videoQA and imageQA state-of-the-art methods, it is difficult to perform an apples-to-apples comparison with majority of these methods as they utilize different computational setups and additional pretraining objectives. Hence, in the table below, we have compared computational resources and training time of our model with two selected state-of-arts that have provided information on computational resources and complexity in their respective papers or source code. As shown, our model (IPRM) requires lesser GPUs and training time than the state-of-art videoQA method SEVILA-BLIP2, and has a higher inference speed (i.e. more samples are processed per second). Similarly for imageQA, we compare IPRM with a replication of the 6-layer concat attention transformer decoder used in MDETR, and find IPRM has marginally lesser peak GPU memory consumption and has comparable training time, while having a marginally higher inference speed.
>
> | Model | GPUs | Total Training time | Inference speed (sample/sec.) |
> | ------- | ------- | ------------------------- | ---------------------------------------- |
> | SEVILA-BLIP2 | 4 x 48GB (NVIDIA A6000) | 8 hours | 0.31 |
> | IPRM | 1 x 46 GB (NVIDIA A40) | 6 hours | 1.11 |
>
>
>
>
> | Model | Peak memory consumption | Total Training Time | Inference speed (samples/sec.) |
> | ------- | ------- | ------------------------- | ---------------------------------------- |
> | Concat-Att-6Layers (MDETR-configuration replication) | 37GB | 10 hours | 192 |
> | IPRM |  33GB | 10 hours | 211 |
>
>
>
>  **Sensitivity analysis of IPRM to hyperparameters**:
> We have provided ablative analysis of IPRM in fig. 6 page 8 of the paper with sensitivity analysis on four primary hyperparameters. These include: i) impact of number of iterative steps (T) and parallel operations (N_op), ii) impact of operation composition unit, iii) impact of reduction ratio and iv) impact of window length . In relation to your suggestion, the first subfigure of fig.6 reports model performances for different combinations of number of iterative steps and parallel operations (with values for each drawn from {1,3,6,9}). Notably, we find that a higher number of iterative steps and parallel operations in IPRM lead to higher performances.

---

> > ### Comment · Reviewer_3tYc · 2024-08-10
> >
> > Thank you for your detailed rebuttal, as well as the places that I missed in the first read. I like the paper so I would like to keep my acceptance rating.

---

> ### Author Response · Authors · 2024-08-12
>
> Dear reviewer 3tYc,
>
> Thank you once again for the useful suggestions and for keeping your acceptance rating.

---

### Official Review · Reviewer_nzfV · 2024-07-13

**Soundness:** 3
**Presentation:** 3
**Contribution:** 3
**Rating:** 7
**Confidence:** 3

**Summary:**

The paper proposes an iterative and parallel reasoning mechanism (IPRM) for VQA tasks, combining step by step iterative computation with parallel processing of independent operations, and also maintaining an internal memory of operation states and results states. Firstly at each reasoning step, previous operation states attend to language features to form latent operations, then latent results are formed by attending to visual features based on the newly formed latent operations and prior result states. Finally new memory state is computed by combining the latent operation and result states alongwith prior memory states with a lookback window. IPRM outperforms previous methods on several complex image and video question answering tasks. IPRM is also lightweight and sample efficient. Ablation experiments demonstrate the importance of different components.

**Strengths:**

1. The paper is well written and easy to follow.
2. A novel way of combining iterative and parallel processing for VQA tasks.
3. The approach demonstrates decent improvements on three video (STAR, AGQAv2, CLEVERER-Humans) and three image question answering tasks (CLVER-Humans, CLEVR-CoGenT, and GQA), alongwith the capability to generalize zeroshot and in out of domain settings (CLEVR-CoGenT).
4. The approach also requires fewer training samples and parameters.

**Weaknesses:**

No major weaknesses, please see questions below.

**Questions:**

1. What is the reason behind applying a MLP with a nonlinear activation instead of just linearly transforming prior operations to get query in Eq 4, as is done in standard attention mechanisms?

2. Can the authors perform some ablations on operation execution component, regarding the importance of using previous result states and newly formed operations in forming attention key?

3. Based on the ablation plots, figure 6 (iii) it seems r=1 performs better than r=2, then why was r=2 chosen for the experiments?

4. For the window length ablation it would be informative to perform the ablation for W>3, to make sure the performance doesn’t increase further with window length. Also it seems W=3 performs slightly better than W=2?

5. How does IPRM do on the CLOSURE [1] dataset (both zeroshot and finetuned)?

Suggestions:

1. It would be informative to also include the MAC model in the comparison of the parameters and sample efficiency in Figure 5.

2. Some of the recent prior work on visual reasoning [2,3,4] haven’t been cited.

3. The authors should include a discussion of limitations/future directions in the main paper.

[1] - Bahdanau, D., de Vries, H., O'Donnell, T.J., Murty, S., Beaudoin, P., Bengio, Y. and Courville, A., 2019. Closure: Assessing systematic generalization of clevr models. arXiv preprint arXiv:1912.05783.

[2] - Mondal, S.S., Webb, T. and Cohen, J.D., 2023. Learning to reason over visual objects. arXiv preprint arXiv:2303.02260.

[3] - Webb, T., Mondal, S.S. and Cohen, J.D., 2024. Systematic visual reasoning through object-centric relational abstraction. Advances in Neural Information Processing Systems, 36.

[4] - Mondal, S.S., Cohen, J.D. and Webb, T.W., 2024. Slot abstractors: Toward scalable abstract visual reasoning. arXiv preprint arXiv:2403.03458.

**Limitations:**

The authors have discussed the limitations in the appendix.

---

> ### Author Rebuttal · Authors · 2024-08-07
>
> Thank you for your thorough review, questions and suggestions. We provide point-by-point clarifications below:
>
> **Reason behind applying a MLP with nonlinear activation to get query in eq. 4**:
> In applying a nonlinear activation for eq.4, we drew inspiration from the cross-attention transformer architecture. In the cross-attention transformer, after each attention block, an MLP is applied which comprises nonlinearities on top of linear projection to model a nonlinear transformation. The output of this MLP then becomes the new ‘query’ for the next transformer block. Hence, for our method, the MLP in eq.4 similarly utilizes a nonlinearity to model a nonlinear transformation and obtain more nuanced ‘queries’ for operation formation at each iterative step. We have also ran an experiment to compare performance without any nonlinearity, and as shown in the table below the next paragraph, IPRM-no-nonlin results in a 1.2% lesser accuracy compared to the original model.
>
>
> **Ablations on operation execution component:**
> We have done two further ablations to analyze the impact of using previous result states (M_{res,t}) and newly formed operations (Z{op,t}) in forming the attention key for operation execution. As shown below, we find that using only M_{res,t} (w/o new-form-ops) results in 0.4% drop while using only Z{op,t} (w/o prev-res-state) results in a 1.1% drop suggesting the previous result states M_{res,t} are more informative in forming the attention key. We will include these results in an elaborated ablations section in the appendix of our updated paper.
>
> | Model | Acc. (%) |
> | --- | --- |
> | IPRM-original | 82.3 |
> | w/o op-form-nonlin | 81.1 |
> | w/o new-form-ops-key | 81.9 |
> | w/o prev-res-state-key | 81.2 |
>
>
> **Why choose r=2 when r=1 performs better?**
> While r=1 does perform better than r=2 by 0.3% accuracy, it requires 0.9M more parameters and 14.5GFLOPs more compute than r=2. Hence, we had chosen r=2 for our optimal model configuration given its parameter and computational efficiency. We will elaborate on this reason in L267 of the updated paper version. We have additionally also reported the breakdown in number of parameters and GFLOPs for different configurations of our model in rebuttal pdf table 1 and will also include it in the appendix of our updated paper.
>
> **Results for window size (W) > 3 and W=3 performs better than W=2?**
> While W=3 does perform marginally better (0.1%) than W=2, similar to the above point, we adopted W=2 as our optimal model configuration as it required 0.9 GFLOPs less compute than W=3. We additionally ran a further experiment with W=4, and found it again does only marginally better (0.13%) better than W=2 while requiring 1.7 GFLOPs more.
>
> **Performance on CLOSURE:**
> We have reported zero-shot results of our model IPRM on CLOSURE below. Note, similar to previous methods (e.g. MAC, FILM, NS-VQA, etc) on this benchmark, we find high standard deviation values ranging from 6% to 8% (currently with 2 random seeds) unlike observed in our experiments with other datasets (where deviation was found to be generally below 0.7%). Further, we have not been able to run finetuned results as of now as the dataset repository does not provide the used finetuning set. IPRM outperforms prior methods on the and_mat_spa, embed_spa_mat and or_mat splits but particularly lags behind on compare_mat and compare_mat_spa splits.
>
> | Model (ZeroShot) | and_mat_spa | embed_mat_spa | embed_spa_mat | or_mat | or_mat_spa | compare_mat | compare_mat_spa |
> | ------- | ------- | ------- | ------- | ------- | ------- | ------- | ------- |
> | IPRM | 83.1±6.4 | 80.3±7.2 | 99.2±0.1 | 75.7±9.4 | 67.1±8.1|  62.2±7.3 | 61.5±5.8 |
> | FiLM | 41.4±18 | 62±11 | 93.4±2 | 34.4±11 | 35.2±5.5 | 66.2±8.5 | 65.8±5 |
> | MAC | 63.7±25 | 76.8±11 | 99±0.15 | 75.2±13 | 70.4±15 | 65.3±8.6 | 66.2±6.4 |
> | NS-VQA* | 33.3±18 | 97.3±4.6 | 98.3±5 | 69.5±28 | 50.4±27 | 96.1±6.2 | 95.4±6.7 |
> | PG-Tensor-NMN* | 37.6±17 | 79.8±1.4 | 95.6±5.6 | 34.1±8.1 | 25.5±11 | 89.2±2.3 | 89.8±2.7 |
> | PG-Vector-NMN* | 32.5±17 | 97.3±2.4 | 97.2±3.9 | 64.9±25 | 47.4±23 | 95.3±5.7 | 94.4±6.4 |
>
>
> *Methods use annotated programs during training for program parser
>
>
> **Include MAC in comparison of parameters and sample efficiency.**
> We have added MAC model in rebuttal pdf fig. 1 (bottom right) for comparison and will update the fig.5 of main paper.
>
> **The authors should include a discussion of limitations/future directions in the main paper.**
> Thank you for mentioning these works on visual reasoning. We will cite and discuss them in the related work section of our updated paper. We will also move the section on limitations/future directions from appendix to the main paper in our updated version.

---

> > ### Comment · Reviewer_nzfV · 2024-08-09
> > **Official comment by Reviewer nzfV**
> >
> > Thank you for the detailed rebuttal and the additional experiments. Can the authors do some failure mode analysis and provide some intuition why IPRM doesn't perform so well on the or_mat and compare_mat settings of the CLOSURE dataset?

---

> ### Author Response · Authors · 2024-08-12
> **Official comment (1/2)**
>
> Dear reviewer nzfV,
>
> As requested, we have conducted an error analysis on the *compare_mat* and *or_mat* settings of CLOSURE dataset by visualizing the intermediate attentions of the model and also analyzing the output predictions. Please note that for the ‘or_mat’ split IPRM achieves the highest performance amongst existing methods. For ‘or_mat_spa’ split IPRM achieves the second-highest performance behind the MAC method (although MAC has a much higher standard deviation of ± 15).
>
> On both the **compare_mat** and **compare_mat_spa** splits, our error analysis suggests that in many cases, IPRM uses the incorrect referent for ‘it’ to perform the ‘comparison’ operation in the second clause.
> * E.g. given the 'compare_mat' question: *“There is another large metallic object that is the same shape as the big green shiny thing; does it have the same color as the tiny matte object?”*, based on its intermediate visual attention, the model appears to correctly perform and identify the first clause object (*‘large metallic object with same shape as big green shiny thing’*). However, in performing the second clause (*‘does it have same color as ..’*), the model appears to use *‘big green shiny thing’* as the referent for *‘it’* rather than the result of the first clause.
> * For more clarity, we have elaborated some cases of this behavior in the comment (2/2) below.
>
> On the **or_mat** and **or_mat_spa** splits (which have numeric counting output labels), our error analysis suggests that IPRM includes the ‘comparison’ object in its count, while the ground truth count does not include it.
> * E.g. given the ‘or_mat’ question: *'How many things are either large purple metallic blocks or large things that are the same shape as the large yellow metal object?'*, the model appears to identify i) large_purple_metal_cube, ii) large_metal_blue_cube  and iii) large_yellow_metal_cube (i.e. the comparison object *'large yellow metal object'* itself) . As a result, its prediction is 3 while ground truth is 2 (which does not include the ‘comparison’ object large_yellow_metal_cube).
> * Please see the elaborated cases in the following comment (2/2) for more clarity.

---

> ### Author Response · Authors · 2024-08-12
> **Official comment (2/2)**
>
> ## Elaborated examples for ‘compare’: ##
> _We use a list of objects instead of the image as we cannot update the rebuttal pdf / upload images._
> ### Example 1 ###
>
> **Question**: there is another large metallic object that is the same shape as the big green shiny thing; does it have the same color as the tiny matte object?
>
> **GT**: no **Pred**: yes
>
> **Objects in scene**:
> - small_rubber_green_cube -> (*‘tiny matte object’*)
> - large_rubber_cyan_cylinder
> - large_metal_brown_cube -> (*‘another large metallic object’*)
> - large_rubber_brown_cylinder
> - large_metal_blue_cylinder
> - large_metal_green_cube -> (*‘big green shiny thing’*)
> - large_rubber_blue_sphere'
>
>
> **Remark**: The model appears to use large_metal_green_cube (i.e. the *‘big green shiny thing’*) and not 'large_metal_brown_cube' to perform the second clause (*‘same color as tiny matte object’*, i.e. small_rubber_green_cube) resulting in prediction of ‘Yes’ (same color) rather than ‘No’.
>
> ### Example 2 ###
> **Question**: there is another yellow metallic object that is the same size as the yellow metal sphere; does it have the same shape as the red matte object?
>
> **GT**: yes **Pred**: no
>
> **Objects in scene:**
> - small_metal_gray_cube
> - large_rubber_red_cube  -> (*‘red matte object’*)
> - large_metal_yellow_sphere
> - large_metal_yellow_cube -> (*‘another yellow metallic object’*)
>
>
> **Remark:** Model uses *“yellow metal sphere”* instead of “large_metal_yellow_cube” to perform *‘same shape’* comparison with 'large_rubber_red_cube'. This results in prediction of ‘No’ (not same shape).
>
> ### Example 3 ###
> **Question:** there is another small gray thing that is the same material as the big yellow sphere; does it have the same shape as the large metallic object?
>
> **GT:** no **Pred:** yes
>
> **Objects in scene:**
> - small_metal_cyan_sphere
> - small_rubber_gray_sphere
> - large_metal_yellow_sphere -> (*‘big yellow sphere’*)
> - small_metal_gray_cube -> (*‘another small gray thing’*)
>
>
> **Remark:** Model uses *“yellow metal sphere”* instead of 'small_metal_gray_cube' to perform *‘same shape’* comparison with 'large_metal_yellow_sphere' (i.e. the *“yellow metal sphere”* itself); results in prediction of ‘Yes’ (same shape) instead of ‘No’.
>
>
>
> ##  Elaborated examples for ‘or_mat’: ##
> _We use a list of objects instead of the image as we cannot update the rebuttal pdf / upload images._
>
> ### Example 1 ###
> **Question:** how many things are either large purple metallic blocks or large things that are the same shape as the large yellow metal object?
>
> **GT**: 2   **Pred**: 3
>
> **Objects in scene:**
> - small_rubber_brown_sphere
> - small_rubber_cyan_cylinder
> - large_rubber_yellow_cylinder
> - small_rubber_purple_cube
> - large_metal_yellow_cube -> (*'large things that are same shape ..'*)
> - large_rubber_yellow_sphere
> - large_metal_blue_cube -> (*'large things that are same shape ..'*)
> - large_metal_purple_cube -> (*‘large purple metallic block’*)
>
> **Remark:** In counting objects with with *‘same shape’* as *‘large yellow metal object’*, the model includes 'large_metal_yellow_cube' (i.e. the 'large yellow metal' itself), resulting in a prediction of 3 total objects (instead of 2 as in ground truth).
>
>
> ### Example 2 ###
> **Question:** how many things are either small gray metallic balls or shiny things that are the same shape as the small purple metal object?
>
> **GT**: 1 **Pred**: 2
>
> **Objects in scene**:
> - small_metal_brown_sphere
> - small_metal_gray_sphere -> (*'small gray metallic balls'*)
> - small_rubber_green_cylinder
> - large_rubber_blue_sphere
> - small_metal_purple_cube -> (*'small purple metal object'*)
> - large_metal_gray_sphere
>
> **Remark**: For the clause *"shiny things that are the same shape as the small purple metal object"*,  the model counts 'small_metal_purple_cube' (i.e. *"small purple metal object"* itself). As a result, its prediction is 2 (including small_metal_gray_sphere) instead of ground truth 1.

---

> ### Comment · Reviewer_nzfV · 2024-08-12
> **Thank you!**
>
> Thank you for the detailed error analysis alongwith the examples for the CLOSURE dataset. It would be good to include the results on CLOSURE and some failure mode analysis in the final version of the main paper. The rebuttal has addressed my concerns. I have increased my rating to 7.

---

> ### Author Response · Authors · 2024-08-12
> **Thank you for your support!**
>
> Thank you once again for your constructive feedback and useful suggestions, and for raising your score. We will include the CLOSURE results and failure mode analysis in the updated version of our paper.

---

### Official Review · Reviewer_oSE3 · 2024-07-13

**Soundness:** 3
**Presentation:** 2
**Contribution:** 3
**Rating:** 5
**Confidence:** 4

**Summary:**

This paper introduces the Iterative and Parallel Reasoning Mechanism (IPRM), a neural architecture for complex visual reasoning and question answering tasks. IPRM involves a three-step iterative process: Operation Formation, Operation Execution, and Operation Composition. In Operation Formation, IPRM generates new parallel operations by extracting relevant language information based on previous operation states. During Operation Execution, these new operations are processed simultaneously, retrieving pertinent visual information guided by both newly formed operations and existing result states. Finally, in Operation Composition, IPRM integrates the new operations and their results into memory by dynamically combining them with each other and with previous operation states.  Experiments are performed on video reasoning like STAR, AGQAv2, visual reasoning datasets like GQA, CLEVR.

**Strengths:**

-	The motivation behind the architecture is interesting and sound.

-	The experiments show improved results comparing with other methods on several benchmarks.

-	The visualization is interesting and somehow can explain intuition of the model.

**Weaknesses:**

•	Some notations need further clarifications: The paper needs to improve consistency and clarity in its notation. For example, it is unclear if K in Figure 2 and its caption is equivalent to N_op used elsewhere. Some variables in equations are not properly defined or explained, such as U and H in equation (9), and u2 and v2 in equations (15) and (16).

•	Model scaling:  How does the performance change if the number of parameters is increased/decreased? While the authors claim IPRM can be applied to different vision and language backbones, they don't provide evidence of its performance when scaling up. This omission leaves questions about the model's scalability and efficiency unanswered.

•	What is the intuition behind using fixed numbers of parallel operators and computation steps, given that different questions likely require different numbers of operations and reasoning steps? Is there any potential drawback of using fixed parameters in this context?

•	Can the author provide some analysis about failure cases of model?

**Questions:**

Please see weaknesses.

---

> ### Author Rebuttal · Authors · 2024-08-07
>
> Thank you for your careful review and questions. We provide point-by-point clarifications below:
>
> **Some notations need further clarifications:**
> Thank you for pointing these out and we apologize for the resulting confusion.
> - Yes, K in fig.2 is equivalent to N_op (which denotes number of parallel operations) used elsewhere. We will use N_op in fig. 2 (and its caption) in our updated paper version to avoid any confusion.
> - U and H in eq.(9) and u2 and v2 in eqs. (15) and (16) are simply used to specify different unique transformation weights and do not denote any variable by themselves. For clarity, in our updated paper, we will use W_{opU} and W_{opH} in eq.(9) (instead of W_{op,U}, W_{op,H}) and W_{opU’} and W_{resV’} in eqs. (15) and (16).
>
> **Model scaling and performance change when number of parameters is increased/decreased. Concerns on model’s scalability and efficiency when scaling up.**
> - Currently, in our ablative analysis in paper section 3.3 fig. 6, we studied four primary factors influencing our method IPRM’s performance and its resultant scale (in terms of parameter size and computational FLOPs). These are: i) number of iterative steps (T), ii) number of parallel operations (N_op), iii) reduction ratio (r), and iv) window length (W). As shown in fig.6, these factors have significant bearing on IPRM’s performance and more capable/higher-scale model variants are obtained as T, N_op and W are increased and r is decreased. E.g. a model with T=1,N_op=1 obtains ~ 75% acc. and when scaled to T=9,N_op=9 obtains ~82% (a 7% performance improvement). We have further reported the computational FLOPs and parameter sizes of IPRM when varying   above factors in table 1 of the attached rebuttal pdf. We will also include it in the appendix of our updated paper version for clarity.
> - Further, in appendix section B.3 table 7 we have also investigated the performance increments when IPRM is applied to different scales of backbone CLIP vision-language (VL) models. As shown, applying IPRM on the larger CLIP VIT-L/14 (369M params) backbone results in 3.4% and 4.3% performance increments on GQA and NLVR2 respectively in comparison to the smaller CLIP VIT-B/16 (149M params). Further, IPRM also outperforms scaling of additional Concat-Att or Cross-Att transformer blocks on CLIP while requiring lesser parameters and comparable GFLOPs. We believe these results help better illustrate performance of IPRM at different backbone model scales and also highlights how integrating IPRM with VL backbones such as CLIP can be more effective than adding further transformer blocks in context of visual reasoning tasks.
>
> **Intuition behind using fixed number of parallel operations and computation steps given varying question demands. Potential drawbacks of fixed number of operations.**
> - Thank you for raising this intriguing point. We agree that different questions can require different number of iterative and parallel operations based on complexity. Halting-based adaptive computation methods such as DACT [1], PonderNet [2] and early-exit strategies [3] have been specifically designed for such a capability. These can certainly be also applied to our architecture owing to its iterative computation nature. However, these approaches typically introduce additional learning losses and task-sensitive hyperparameters (such as computational budget or halting penalty) that can vary across tasks and datasets.
>
> - To enable our proposed reasoning model to be uniformly applicable to different image and video reasoning tasks without modification, we opted for weighted residual connections in updating the memory state (eqs. 15 & 16) over fixed steps. This implicitly enables the model to perform identity operations for steps where no additional computation may be required. This can be seen in reasoning visualizations in rebuttal pdf fig.1 and appendix fig. 8 to 11, where the lang. and vis. attention remains minimally changed b/w consecutive iterative steps.
>
> - A potential drawback of such fixed computation is that the amount of GFLOPs remains constant regardless of the question complexity. However, we note that this is true for other baseline reasoning methods and transformer-based models (such as SeViLA-BLIP2, Concat-Att, MIST, and MDETR) as well, and hence can be an interesting direction to explore in future.
>
>
> **Analysis about failure cases of model**
> - Currently, in appendix fig. 10 & 11 and examples in rebuttal pdf fig.1, we provide qualitative analysis for a few representative failure cases of our model. These largely stem from the model either not having sufficient commonsense or the model not utilizing visual information correctly. E.g. in fig. 10, for the question, “Are the two objects that are of a primary color, but not red, of the same shape?”, the model appears to not understand what a “primary color” is, and hence fails in its reasoning. Similarly, in the rebuttal pdf bottom-left example, the model appears to correctly find the ‘device that is on?’, but outputs an imprecise label (‘computer’) instead of the ground truth label (‘monitor’).
>
> - We have also performed quantitative analysis on the impact of visual encoding quality on our model’s performance in appendix table 5 (left). As shown, when utilizing ground-truth visual inputs, our model can achieve a 9.4% performance boost on STAR. Similarly on GQA, we observed a 25% performance boost compared to when using predicted visual inputs. This suggests that many errors of our model may stem from imprecise backbone visual outputs rather than the reasoning process itself, and can thereby be improved with advancements in visual backbones.
>
> [1] Eyzaguirre, Cristobal, and Alvaro Soto. "Differentiable adaptive computation time for visual reasoning." CVPR. 2020.
>
> [2] Banino, Andrea, Jan Balaguer, and Charles Blundell. "Pondernet: Learning to ponder." arXiv preprint arXiv:2107.05407 (2021).
>
> [3] Guo, Qiushan, et al. "Dynamic recursive neural network."CVPR. 2019.

---

> ### Comment · Area_Chair_eziG · 2024-08-13
> **Reviewer oSE3: please respond to the authors' rebuttal**
>
> Dear reviewer,
>
> thanks for your participation in the NeurIPS peer review process.  We are waiting for your response to the rebuttal.  You gave a borderline rating (5). Is the response from authors satisfactory? Does it address weaknesses that you mentioned in the review?
>
> - If yes, are you planning to increase your score?
> - If no, could you help the authors understand how they can improve their paper in future versions?
>
> Thanks,
> AC

---

### Author Rebuttal · Authors · 2024-08-07

We thank all reviewers for their time in reviewing the paper and for their positive and thoughtful feedback. We are encouraged that they found our proposed architecture and underlying motivation to combine iterative and parallel computation for complex visual reasoning tasks to be novel (nzfV, 3tYc, P4wf), interesting and sound (oSE3, P4wf).

Further, we are pleased that they found the paper to be well written and generally well-structured (P4wf, nzfV, 3tYc) and the experiments to be extensive with decent performance improvements (nzfV, 3tYc, oSE3) across several challenging videoQA and imageQA benchmarks along with an in-depth analysis of the method (P4wf) and how it requires fewer training samples and parameters (nzfV).

We have addressed the concerns raised by the reviewers pointwise. Some **primary concerns** are highlighted below with a brief description of our response:

**Reviewer oSE3** -- We have i) addressed concerns regarding _scalability of the model and efficiency of the model when scaling up_ (with model ablations and results using different CLIP backbones), ii) provided intuition behind using fixed number of parallel operations and computation steps given varying question demands, and iii) provided analysis about failure cases of model. We have also clarified some confusion regarding the notation which we will rectify in the updated paper.

**Reviewer nZFw** -- We have added _zero-shot results of IPRM on CLOSURE dataset_ where IPRM demonstrates superior performance on multiple dataset splits. We have also clarified our motivation behind query formation, reported additional ablations regarding 'key formation' in operation execution, and included the MAC model in comparison of parameters and sample efficiency of visual reasoning methods.

**Reviewer 3tYc** -- We have provided more extensive details on the scalability of IPRM in terms of computational resources and training time. We have also compared it with baseline transformer modules and state-of-art methods on multiple scalability factors such as peak GPU memory usage, training time and inference speed. Further, we have also specified ablations related to sensitivity analysis of model hyperparameters.

**Reviewer P4wf** -- We have provided directions on how IPRM can be scaled to a video-language model that can provide zero-shot results on multiple video-datasets and support multi-turn conversations given its similarity to conventional transformer modules in terms of parallel processing of inputs and end-to-end modular integration with vision-language models. We also discuss how ideas in SOTA methods such as  “Look, Remember and Reason” (LRR) and “Coarse-to-Fine Reasoning” (CFR)  can be used to make IPRM even better and provide an experiment to validate one such idea based on usage of symbolic scene graph inputs. Further, we have demonstrated language and visual attention visualization on a real-world dataset i.e. GQA, and provided clarifications regarding model interpretability visualization for paper fig. 7.

We hope we have addressed the major questions and concerns of reviewers, and look forward to further discussion.

---

### Decision · Program_Chairs · 2024-09-25

**Decision:**

Accept (poster)

**Comment:**

Learning to

Summary of Review Process:
- 4 reviews.
- Leaning positive scores: 5,6,7,7
- Authors submitted rebuttals/responses
- 1 of 4 reviewers did not respond to rebuttal

Meta Review:
In this work, LLMs and image generation models are leveraged to create synthetic image-text pairs for VLM training. An LLM generates captions and a T2I model uses these captions to generate synthetic images.

Reviewers appreciated the following points about this paper:
- novelty of the framework, especially combining iterative and parallel reasoning processes
- extensive experiments and clear demonstration of performance improvement
- well written / well structured manuscript

Reviewer oSE3 asked about scalbility and efficiency, intution behind number of parallel operators, failure analysis, and notation confusion. The reviewer did not acknowledge the authors' response. The AC has read the responses and thinks that these address the questions. The AC recommends that it would be important to integrate these responses into the final version to improve readibility and the flow of the paper

Reviewer nZFw asked about the motivation behind using an MLP with nonlinear activattion, additional ablations, additional results on CLOSURE, additional baseline MAC, error analysis, and limitations section.  The reviewer deemed the authors' responses satisfactory and increased the rating to 7 (Accept).

Reviewer 3tYc suggested some improvements to the structure of the paper and a detailed discussion about IPRM, including scalability and sensitivity analysis.

Reviewer P4wf asked about generalization on multiple video dataset, multiturn conversations, comparison to recent work and improvements to the design of Fig 7.

AC: There is clear support from all reviewers to accept this paper. The questions raised by the reviewers were important and the authors were able to address those during the discussion phase. I strongly encourage the authors to incorporate these changes and integrate clarifications and discussions into the final paper.